# Understanding Contrastive Learning through Variational Analysis and Neural Network Optimization Perspectives

## Abstract

The SimCLR method for contrastive learning of invariant visual representations has become extensively used in supervised, semi-supervised, and unsupervised settings, due to its ability to uncover patterns and structures in image data that are not directly present in the pixel representations. However, the reason for this success is not well-explained, since it is not guaranteed by invariance alone. In this paper, we conduct a mathematical analysis of the SimCLR method with the goal of better understanding the geometric properties of the learned latent distribution. Our findings reveal two things: (1) the SimCLR loss alone is not sufficient to select a *good* minimizer — there are minimizers that give trivial latent distributions, even when the original data is highly clustered — and (2) in order to understand the success of contrastive learning methods like SimCLR, it is necessary to analyze the neural network training dynamics induced by minimizing a contrastive learning loss. Our preliminary analysis for a one-hidden layer neural network shows that clustering structure can present itself for a substantial period of time during training, even if it eventually converges to a trivial minimizer. To substantiate our theoretical insights, we present numerical results that confirm our theoretical predictions.

## 1 Introduction

Unsupervised learning of effective representations for data is one of the most fundamental problems in machine learning, especially in the context of image data. The widely successful *discriminative approach* to learning representations of data is most similar to fully supervised learning, where features are extracted by a backbone convolutional neural network, except that the fully supervised task is replaced by an unsupervised or *self-supervised* task that can be completed without labeled training data.

Many successful discriminative representation learning methods are based around the idea of finding a feature map that is *invariant* to a set of transformations (i.e., data augmentations) that are expected to be present in the data. For image data, the transformations may include image scaling, rotation, cropping, color jitter, Gaussian blurring, and adding noise, though the question of which augmentations give the best features is not trivial (Tian et al., 2020). Invariant feature learning methods include VICReg Bardes et al. (2021), Bootstrap Your Own Latent (BYOL) (Grill et al., 2020), Siamese neural networks Chicco (2021), and contrastive learning techniques such as SimCLR Chen et al. (2020) (see also (Hadsell et al., 2006; Dosovitskiy et al., 2014; Oord et al., 2018; Bachman et al., 2019)).

In contrastive learning, the primary self-supervised task is to differentiate between positive and negative pairs of data instances. The goal is to find a feature map for which positive pairs have maximally similar features, while negative pairs have maximal different features. The positive and negative examples do not necessarily correspond to classes. In SimCLR, positive pairs are images that are the same up to a transformation, while all other pairs are negative pairs. Contrastive learning has also been successfully applied in supervised (Khosla et al., 2020) and semi-supervised contexts (Li et al., 2021; Yang et al., 2022; Singh, 2021; Zhang et al., 2022b; Lee et al., 2022; Kim et al.,

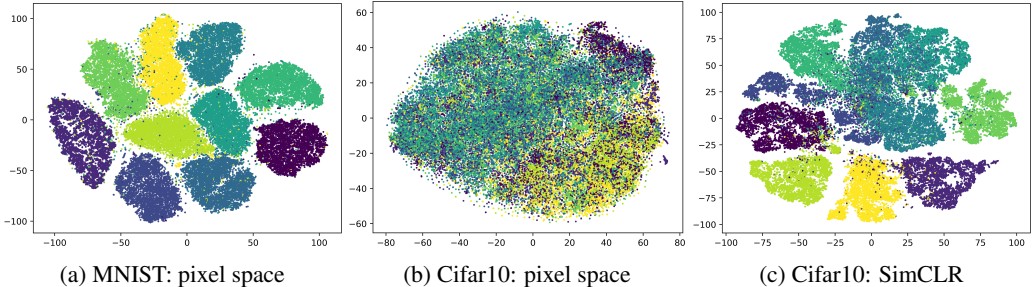

(a) MNIST: pixel space          (b) Cifar10: pixel space          (c) Cifar10: SimCLR

Figure 1: t-SNE visualizations of the MNIST and Cifar10 data sets. In (a) and (b) the images are represented by the raw pixels, while (c) gives a visualization of the SimCLR embedding. This illustrates how SimCLR is able to uncover clustering structure in data sets.

2021; Ji et al., 2023), and has been used for learning Lie Symmetries of partial differential equations Mialon et al. (2023) (for a survey see Le-Khac et al. (2020)).

All invariance based feature extraction techniques must address the fundamental problem of dimension collapse, whereby a method learns the trivial constant map $f(x) = c$ (or a very low rank map), which is invariant to *all* transformations, but not informative or descriptive. There are various ways to prevent dimension collapse. In contrastive learning the role of the negative pairs is to prevent collapse by creating repulsion terms in the latent space, however, full or partial collapse can still occur (Jing et al., 2021; Zhang et al., 2022a; Shen et al., 2022; Li et al., 2022). In BYOL collapse is prevented by halting backpropagation in certain parts of the loss, and incorporating temporal averaging. In VICReg, additional terms are added to the loss function to maintain variance in each latent dimension, as well as to decorrelate variables.

Provided dimensional collapse does not occur, a fundamental unresolved question surrounding many feature learning methods is: why do they work so well at producing embeddings that uncover key features and patterns in data sets? As a simple example, consider fig. 1. In fig. 1a and fig. 1b we show t-SNE (Van der Maaten & Hinton, 2008) visualizations of the MNIST (Deng, 2012) and Cifar-10 (Krizhevsky et al., 2009) data sets, respectively, using their pixel representations. We can see that visual features are not required on MNIST, which is highly preprocessed, while for Cifar-10 the pixel representations are largely uninformative, and feature representations are essential. In fig. 1c we show a t-SNE visualization of the latent embedding of the SimCLR method applied to Cifar-10, which indicates that SimCLR has uncovered a strong clustering structure in Cifar-10 that was not present in the pixel representation.

The goal of this paper is to provide a framework that can begin to address this question, and in particular, to explain fig. 1. To do this, we assume the data follows a *corruption* model, where the observed data is derived from some clean data with distribution $\mu$ that is highly structured or clustered in some way (e.g., follows the manifold assumption with a clustered density). The observed data is then obtained by applying transformations at random from a set of augmentations $\mathcal{T}$ to the clean data points (i.e., taking different views of the data), producing a corrupted distribution $\widetilde{\mu}$. The main question that motivated our work is that of understanding what properties of the original clean data distribution $\mu$ can be uncovered by unsupervised contrastive feature learning techniques? That is, once an invariant feature map $f : \mathbb{R}^D \to \mathbb{R}^d$ is learned, is the latent distribution $f_{\#}\widetilde{\mu}$ similar in any to the clean distribution $\mu$, or can it be used to deduce any geometric or topological properties of $\mu$?

This paper has two main contributions. For simplicity we focus on SimCLR, and indicate in the appendix how our results extend to other techniques.

1. We show minimizing the SimCLR contrastive learning loss is not sufficient to recover information about $\mu$. In particular, there are invariant minimizers of the SimCLR loss that are completely independent of the data distributions $\mu$ and $\widetilde{\mu}$. In the extreme case, the original clean data may be highly clustered, while the latent distribution has a minimizer that is the uniform distribution.

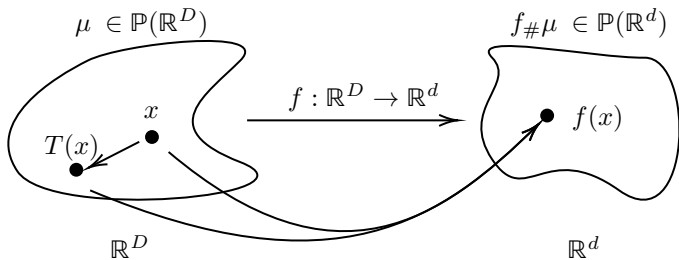

Figure 2: Illustration of an invariant feature map $f : \mathbb{R}^D \to \mathbb{R}^d$ that maps the data distribution $\mu$ to the feature distribution $f_{\#}\mu$ in the latent space, along with a perturbation function $T : \mathbb{R}^D \to \mathbb{R}^D$. The figure shows that both the original point $x$ and the perturbed point $T(x)$ map to $f(x)$ in the feature space.

2. To understand the success of contrastive learning, it is necessary to analyze the neural network training dynamics induced by gradient descent on the SimCLR loss. Using the neural kernel approach, we show that clusterability structures in $\mu$ strongly affect the training dynamics and can remain present in the latent distribution for a long time, even if gradient descent converges to a trivial minimizer.

Our work complements research on dimension collapse in contrastive learning (Jing et al., 2021; Zhang et al., 2022a; Shen et al., 2022; Li et al., 2022), as our findings hold even without collapse. We also highlight recent work (Meng & Wang, 2024) on the training dynamics of contrastive learning through a continuum limit PDE. Other related works, such as (HaoChen et al., 2021; Balestriero & LeCun, 2022), provide guarantees for downstream tasks like semi-supervised learning by studying the alignment between class-membership clusters and an "augmentation graph." Our paper complements these by examining when this alignment holds in contrastive learning.

**Outline:** In section 2 we overview contrastive learning, and our corruption model for the data. In section 3 we derive and study the optimality conditions for the SimCLR loss, and give conditions for stationary points. In section 4 we study the neural dynamics of training SimCLR for a one-hidden layer neural network.

## 2 CONTRASTIVE LEARNING

We describe here our model for corrupted data in the setting of contrastive learning, and a reformulation of the SimCLR loss that is useful or our analysis. Let $\mu \in \mathbb{P}(\mathbb{R}^D)$ be a data distribution in $\mathbb{R}^D$. Let $\mathcal{T}$ be a set of transformation functions $T : \mathbb{R}^D \to \mathbb{R}^D$ that is measurable such that, for a given $x \in \mathbb{R}^D$, $T(x) \in \mathbb{R}^D$ represents a perturbation of $x$, such as a data augmentation (e.g., cropping and image, etc.). Let $\widetilde{\mu} \in \mathbb{P}(\mathbb{R}^D)$ denote the distribution obtained by perturbing $\mu$ with the perturbations defined in $\mathcal{T}$. That is, we choose a probability distribution $\nu \in \mathbb{P}(\mathcal{T})$ over the perturbations, and samples from $\widetilde{\mu}$ are generated by sampling $x \sim \mu$ and $f \sim \nu$, and taking the composition $f(x)$.

We treat $\mu$ as the original clean data, which is not observable, while the perturbed distribution $\widetilde{\mu}$ is how the data is presented. Our goal is to understand whether contrastive learning can recover information about the original data $\mu$, provided the distribution of augmentations $\nu$ is known.

Ostensibly, the objective of contrastive learning is to identify an embedding function $f : \mathbb{R}^D \to \mathbb{R}^d$ that is invariant to the set of transformations $\mathcal{T}$. Provided such an invariant map is identified, $f$ pushes forwards both $\mu$ and $\widetilde{\mu}$ to the same latent distributions, that is

$$f_{\#}\widetilde{\mu} = f_{\#}\mu.$$

As a result, the desirable map $f$ is not only invariant to perturbations from $\mathcal{T}$ but also successfully retrieves the unperturbed data $\mu$, ensuring that the embedded distribution $f_{\#}\mu$ serves as a pure feature representation of the given data. However, it is far from clear how $\mu$ and $f_{\#}\mu$ are related, and whether any interesting structures in $\mu$ (such as clusterability) are also present in $f_{\#}\mu$.

For instance, if $\widetilde{\mu}$ represents image data, contrastive learning aims to discover a feature distribution $f_{\#}\widetilde{\mu}$ that remains invariant to transformations such as random translation, rotation, cropping, Gaus-

sian blurring, and others. Figure 2 illustrates the mapping $f : \mathbb{R}^D \to \mathbb{R}^d$ and $T \in \mathcal{T}$. As a result, this feature distribution effectively captures the essential characteristics of the data without being influenced by these perturbations. These feature distributions are often leveraged in downstream tasks such as classification, clustering, object detection, and retrieval, where they achieve state-of-the-art performance (Le-Khac et al., 2020).

To achieve this, a cost function is designed to bring similar points closer and push dissimilar points apart through the embedding map, using attraction and repulsion forces. A popular example is the Normalized Temperature-Scaled Cross-Entropy Loss (NT-Xent loss) introduced by Chen et al. (2020), which leads to the optimization problem

$$\min_{f:\mathbb{R}^D \to \mathbb{R}^d} \mathbb{E}_{x \sim \mu, T, T' \sim \nu} \log \left( 1 + \frac{\sum_{h \in \{T, T'\}} \mathbb{E}_{y \sim \mu} \left[ \mathbb{1}_{x \neq y} \exp \left( \frac{\mathrm{sim}_f(T(x), h(y))}{\tau} \right) \right]}{\exp \left( \frac{\mathrm{sim}_f(T(x), T'(x))}{\tau} \right)} \right), \quad (1)$$

where $\nu \in \mathbb{P}(\mathcal{T})$ is a probability distribution on $\mathcal{T}$, which is assumed to be a measurable space, $\tau$ is a given parameter, and $\mathrm{sim}_f : \mathbb{R}^D \times \mathbb{R}^D \to \mathbb{R}$ is a function measuring the similarity between two embedded points with $f$ in $\mathbb{R}^d$ defined as:

$$\mathrm{sim}_f(x, y) = \frac{f(x) \cdot f(y)}{\|f(x)\| \|f(y)\|}. \quad (2)$$

The denominator inside the log function acts as an attraction force between perturbed points from the same sample $x$, while minimizing the numerator acts as a repulsion force between points from different samples $x$ and $y$. Thus, the minimizer $f$ of the cost is expected to exhibit invariance under the group of perturbation functions from $\mathcal{T}$.

$$f(T(x)) = f(x), \qquad \forall\, x \in \mathbb{R}^D, \ \forall\, T \in \mathcal{T}. \quad (3)$$

The repulsion force prevents dimensional collapse, where the map sends every sample to a constant: $f(x) = c$ for all $x \in \mathbb{R}^D$.

Our first observation is that the NT-Xent loss becomes independent of the data distribution once $f$ is invariant, and so the latent distribution for an invariant minimizer may be completely unrelated to the input data.

**Proposition 2.1.** *Suppose $\mu \in \mathbb{P}(\mathbb{R}^d)$ is absolutely continuous and the embedding map $f : \mathbb{R}^D \to \mathbb{R}^d$ is invariant under the distribution $\nu$, satisfying eq. (3). Applying a change of variables, we obtain the following reformulation from eq. (1):*

$$\min_{f:\mathbb{R}^D \to \mathbb{R}^d} \mathbb{E}_{x \sim f_{\#}\mu} \log \left( 1 + 2 \mathbb{E}_{y \sim f_{\#}\mu} \left[ \mathbb{1}_{x \neq y} \exp(\mathrm{sim}_f(x, y)/\tau) \right] \right)$$

$$= \min_{\rho \in \mathbb{P}(\mathbb{R}^d)} \mathbb{E}_{x \sim \rho} \log \left( 1 + 2 \mathbb{E}_{y \sim \rho} \left[ \mathbb{1}_{x \neq y} \exp(\mathrm{sim}(x, y)/\tau) \right] \right),$$

*where $\mathrm{sim}(x, y) = \mathrm{sim}_{\mathrm{Id}}(x, y) = \frac{x \cdot y}{\|x\| \|y\|}$.*

The result in Proposition 2.1 shows that minimizing the NT-Xent cost with respect to an embedding map, once the map is invariant, is equivalent to minimizing over the probability distribution in the latent space. This minimization is *completely independent* of the input data distribution $\mu$. A similar phenomenon is observed in other unsupervised learning models like VICReg (Bardes et al., 2021) and BYOL (Grill et al., 2020), with a similar derivation provided in the appendix.

Next, we will analyze the NT-Xent loss by studying its minimizer and the dynamics of gradient descent. To avoid issues with the nondifferentiability of angular similarity $\mathrm{sim}_f$ and the nonuniqueness of solutions (e.g., $kf$ is also a minimizer for any $k > 0$), we reformulate the loss to simplify the analysis. This leads to the generalized formulation of the NT-Xent loss in equation 1.

**Definition 2.1.** The cost function we consider for contrastive learning is

$$\inf_{f \in \mathcal{C}} \left\{ L(f) := \mathbb{E}_{x \sim \mu, T, T' \sim \nu} \Psi \left( \frac{\mathbb{E}_{y \sim \mu} \eta_f(T(x), T'(y))}{\eta_f(T(x), T'(x))} \right) \right\}, \quad (4)$$

where $\Psi : \mathbb{R} \to \mathbb{R}$ is a nondecreasing function, $\mathcal{C}$ is a constraint set, and $\eta_f$ is defined as

$$\eta_f(x, y) = \eta(\|f(x) - f(y)\|^2 / 2), \quad (5)$$

where $\eta : \mathbb{R}_{\geq 0} \to \mathbb{R}$ is a differentiable similarity function that is maximized at 0.

The formulation in eq. (4) generalizes the original formulation in eq. (1) by removing the indicator function $\mathbb{1}_{x \neq y}$, as the effect of this function becomes negligible when a large $n$ is considered. Furthermore, the generalized formulation introduces a differentaible similarity function. This simplifies the analysis of the minimizer in the variational formulation. The generalized formulation can easily be related to the original cost function in eq. (1) by setting $\Psi(t) = \log(1 + t)$, $\eta(t) = e^{-t/\tau}$ and defining $\mathcal{C} = \{f : \mathbb{R}^D \to \mathbb{S}^{d-1}\}$. Then, the similarity function $\eta_f$ retains the same interpretation as angular similarity $\text{sim}_f$. This is because, if $f$ lies on the unit sphere in $\mathbb{R}^d$, and so

$$\exp\left(-\frac{1}{2\tau}\|f(x) - f(y)\|^2\right) = \exp\left(\frac{1}{\tau}(f(x) \cdot f(y) - 1)\right) = C \exp\left(\frac{1}{\tau} \frac{f(x) \cdot f(y)}{\|f(x)\|\|f(y)\|}\right),$$

where $C = \exp(-1/\tau)$. The consideration of the constraint also resolves the issue in eq. (1), where $kf$, for any $k \in \mathbb{R}$, could be a minimizer of eq. (1). Thus, in the end, the introduced formulation in eq. (4) remains fundamentally consistent with the original NT-Xent cost structure.

## 3 OPTIMALITY CONDITION

In this section, we aim to find the optimality condition for eq. (4) and analyze properties of the minimizers. Our first result provides the first order optimality conditions.

**Proposition 3.1.** *The first optimality condition of the problem eq.* (4) *takes the form*

$$\int_{\mathcal{T}} \int_{\mathbb{R}^D} \left\langle \int_{\mathcal{T}} \int_{\mathbb{R}^D} \left(\frac{\Psi'(G_{T,T'}(f, x))}{\eta_f(T(x), T'(x))} + \frac{\Psi'(G_{T,T'}(f, y))}{\eta_f(T(y), T'(y))}\right) \eta_f'(T(x), T'(y))\left(f(T(x)) - f(T'(y))\right) \right.$$
$$- \left(\Psi'(G_{T,T'}(f, x))\eta_f(T(x), T'(y)) + \Psi'(G_{T',T}(f, x))\eta_f(T'(x), T(y))\right)$$
$$\left. \frac{\eta_f'(T(x), T'(x))}{\eta_f^2(T(x), T'(x))}\left(f(T(x)) - f(T'(x))\right)d\mu(y)d\nu(T'), h(T(x))\right\rangle d\mu(x)d\nu(T) = 0 \quad (6)$$

*for all $h$ such that $f + h \in \mathcal{C}$ where $\eta_f'(x, y) = \eta'(\|f(x) - f(y)\|^2/2)$, $\Psi(t) = \log(1 + t)$ and $G_{T,T'}(f, x) = \frac{\mathbb{E}_{z \sim \mu} \eta_f(T(x), T'(z))}{\eta_f(T(x), T'(x))}$.*

*If $f$ is invariant to the perturbation in $\nu$, then the gradient of $L$ takes the form*

$$\nabla L(f)(x) = \int_{\mathbb{R}^D} \left(\Psi'(G_{\text{Id},\text{Id}}(f, x)) + \Psi'(G_{\text{Id},\text{Id}}(f, y))\right) \eta_f'(x, y)(f(x) - f(y))d\mu(y). \quad (7)$$

Using the first optimality condition described in Proposition 3.1, we can characterize the minimizer of the NT-Xent loss in eq. (4). The following theorem describes the possible local minimizers of eq. (4), considering the constraint set defined as $\mathcal{C} = \{f : \mathbb{R}^D \to \mathbb{S}^{d-1}\}$.

**Theorem 3.2.** *Given a data distribution $\mu \in \mathbb{P}(\mathbb{R}^D)$, let $f \in \mathcal{C} = \{f : \mathbb{R}^D \to \mathbb{S}^{d-1}\}$ be an invariant map such that the embedded distribution $f_\#\mu$ is a symmetric discrete measure satisfying*

$$\int_{\mathbb{S}^{d-1}} h(x_1, y)df_\#\mu(y) = \int_{\mathbb{S}^{d-1}} h(x_2, y)df_\#\mu(y), \quad (8)$$

*for all $x_1, x_2 \sim f_\#\mu$ and for all anti-symmetric functions $h : \mathbb{S}^{d-1} \times \mathbb{S}^{d-1} \to \mathbb{S}^{d-1}$ such that $h(x, y) = -h(y, x)$. Then, $f$ is a stationary point of eq.* (4) *in $\mathcal{C}$.*

*Remark* 3.1. Examples of the embedded distribution $f_\#\mu$ in Theorem 3.2 include a discrete measure, $f_\#\mu = \frac{1}{n}\sum_{i=1}^{n} \delta_{x_i}$, with points $x_i$ evenly distributed on $\mathbb{S}^{d-1}$, or all points mapped to a single point, $f_\#\mu = \{x\}$. Figure 3 shows loss plots for different embedded distributions, $f_\#\mu = \frac{1}{K}\sum_{i=1}^{K} \delta_{x_i}$, with points $x_i$ evenly distributed on $\mathbb{S}^1$, illustrating how each stationary point relates to the loss.

The first plot shows the loss decreasing with the number of clusters, leveling off after a certain point. The second plot shows the loss decreasing as the minimum squared distance between cluster points narrows, plateaus once a threshold is reached. Both suggest that increasing the number of clusters or using a uniform distribution on $\mathbb{S}^1$ minimizes the NT-Xent loss. Additionally, increasing the number of points and decreasing $\tau$ further reduces the loss. The third plot reveals a linear relationship between $\tau$ and the threshold for the minimum squared distance, offering insight into the optimal cluster structure for minimizing the loss at a given $\tau$.

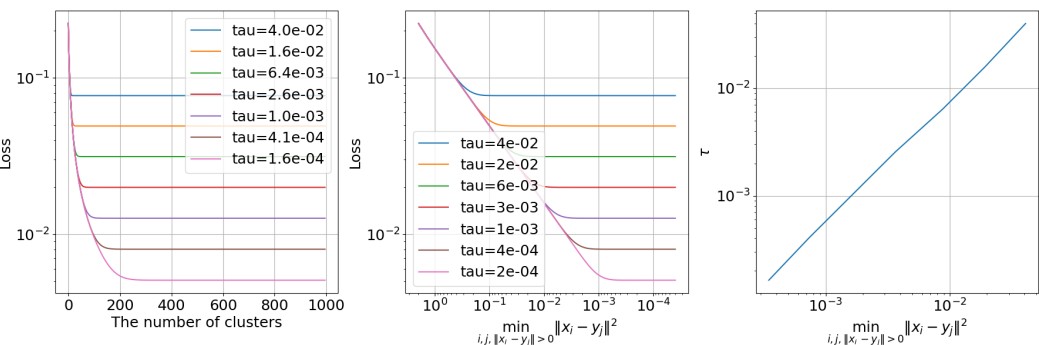

Figure 3: The figure shows the NT-Xent loss for different embedded distributions $f_{\#}\mu = \frac{1}{K}\sum_{i=1}^{K}\delta_{x_i}$ with $x_i$ on $\mathbb{S}^1$. The first plot shows the loss decreasing with the number of clusters, then plateauing. The second shows the loss decreasing with the minimum squared distance between cluster points, stopping at a threshold. Both suggest that increasing clusters and decreasing $\tau$ reduce the loss. The third plot shows a linear relationship between $\tau$ and the minimum distance.

*Remark* 3.2. Theorem 3.2 is related to the result from Wang & Isola (2020), where the authors studied local minimizers by minimizing the repulsive force under the assumption of an invariant feature map. They showed that, asymptotically, the uniform distribution on $\mathbb{S}^{d-1}$ becomes a local minimizer as the number of negative points increases. Our result extends this by offering a more general, both asymptotic and non-asymptotic characterization of local minimizers, broadening their findings.

It follows from Theorem 3.2 that gradient descent on the NT-Xent loss can lead to solutions that are *completely independent* of the original data distribution $\mu$. For instance, if $\mu$ has some underlying cluster structure, with multiple clusters, there are minimizers of the NT-Xent loss, i.e., an invariant map $f$, that map onto an *arbitrary* distribution in the latent space, completely independent of the clustering structure of $\mu$. However, in practice, when the map is parameterized using neural networks, and trained with gradient descent on $L(f)$, it is very often observed that the clustering structure of the original data distribution $\mu$ emerges in the latent space (see fig. 1). In fact, our results in section 4 show that this is true even if we initialize gradient descent very poorly, starting with an invariant $f$ mapping to the uniform distribution $\mathcal{U}(\mathbb{S}^{d-1})$!

Although the contrastive loss $L(f)$ has minimizers that ignore the data distribution $\mu$, leading to poor results, contrastive learning often achieves excellent performance in practice. This suggests that the neural network's parameterization and gradient descent optimization are *selecting* a good minimizer for $L(f)$, producing well-clustered distributions in the latent space. To understand this, we will analyze the dynamics of neural network optimization during training in the following sections.

## 4 OPTIMIZATION OF NEURAL NETWORKS

Here, we study contrastive learning through the lens of the associated neural network training dynamics, which illustrates how the data distribution enters the latent space through the neural kernel. In this section, we use the notation $[\![n]\!] = \{1, \ldots, n\}$.

### 4.1 GRADIENT FLOW FROM NEURAL NETWORK PARAMETERS

Let $w \in \mathbb{R}^m$ be a vector of neural network parameters, $\{x_1, \ldots, x_n\} \subset \mathbb{R}^D$ be data samples, and $f(w, x_i) = (f^1(w, x_i), \ldots, f^d(w, x_i))^\top \in \mathbb{R}^d$ be an embedding function where each function $f^k : \mathbb{R}^{n+D} \to \mathbb{R}$ is a scalar function for $k = 1, \ldots, d$. Consider a loss function $L = L(y^1, \cdots, y^d) : \mathbb{R}^d \to \mathbb{R}$ with respect to $w$:

$$\mathcal{L}(w) = \frac{1}{n}\sum_{i=1}^{n} L(f(w, x_i)). \tag{9}$$

Let $w(t)$ be a vector of neural network parameters as a function of time $t$. The gradient descent flow can be expressed as

$$\dot{w}(t) = -\nabla \mathcal{L}(w).$$

Due to the highly non-convex nature of $\mathcal{L}$, this gradient flow is difficult to analyze. By shifting the focus to the evolution of the neural network's output on the training data over time, rather than the weights, we can derive an alternative gradient flow with better properties for easier analysis. The following proposition outlines this gradient flow derived from the loss function $\mathcal{L}$. The proof of the proposition is provided in the appendix.

**Proposition 4.1.** *Let $w(t)$ be a vector of neural network parameters as a function of time $t$. Consider a set of data samples $\{x_1, \ldots, x_n\}$. Define a matrix function $z : \mathbb{R} \to \mathbb{R}^{d \times n}$ such that*

$$z(t) = [f(w(t), x_1) \quad f(w(t), x_2) \quad \cdots \quad f(w(t), x_n)]. \tag{10}$$

*Let $z_i$ denote the $i$-th column of $z$. Then, $z_i(t)$ satisfies the following ordinary differential equation (ODE) for each $i \in [\![n]\!]$:*

$$\dot{z}_i(t) = -\frac{1}{n} \sum_{j=1}^{n} K_{ij}(t) \nabla L(z_j(t)) \in \mathbb{R}^d, \tag{11}$$

*where the kernel matrix $K_{ij} \in \mathbb{R}^{d \times d}$ is given by*

$$(K_{ij}(t))^{kl} = K_{ij}^{kl}(t) = (\nabla_w f^k(w(t), x_i))^\top (\nabla_w f^l(w(t), x_j)). \tag{12}$$

*Remark* 4.1. We remark that the viewpoint in proposition 4.1, of lifting the training dynamics from the neural network weights to the function space setting, is the same that is taken by the Neural Tangent Kernel (NTK) Jacot et al. (2018). The difference here is that we do not consider an *infinite width* neural network, and we evaluate the kernel function on the training data, so the results are stated with kernel matrices that are data dependent (which is important in what follows). In fact, it is important to note that proposition 4.1 is very general and holds for any parameterization of $f$, e.g., we have so far not used that $f$ is a neural network.

*Remark* 4.2. The training dynamics in the absence of a neural network can be expressed as

$$\dot{z}_i(t) = -\nabla L_i(z_i(t)) \in \mathbb{R}^d \tag{13}$$

where $K_{ij}$ is set to be identity matrices. In contrast to eq. (11), the above expression shows that the training dynamics on the $i$-th point $z_i$ are influenced solely by the gradient of the loss function at $x_i$, and there is no mixing of the data via the neural kernel $K$ (since here it is the identity matrix).

Using proposition 4.1 we can study the invariance-preserving properties (or lack thereof) of gradient descent with and without the neural network kernel.

**Theorem 4.2.** *Consider the gradient descent iteration from a gradient flow without a neural network in eq. (13), where $z_i^{(b)} = f(w^{(b)}, x_i)$ for all $i \in [\![n]\!]$, and*

$$z_i^{(b+1)} = z_i^{(b)} - \sigma \nabla L(z_i^{(b)}), \tag{14}$$

*with $\sigma$ as the step size. If $f(w^{(0)}, \cdot)$ is invariant to perturbations from $\nu$, as defined in eq. (17), then $f(w^{(b)}, \cdot)$ remains invariant for all gradient descent iterations.*

*On the other hand, in the case of a gradient descent iteration from eq. (10),*

$$z_i^{(b+1)} = z_i^{(b)} - \frac{\sigma}{n} \sum_{j=1}^{n} K_{ij}^{(b)} \nabla L(z_j^{(b)}), \tag{15}$$

*the invariance of $f$ at the $(b+1)$-th iteration holds only if $f$ is invariant at the $b$-th iteration and additionally satisfies the condition $\nabla_w f(w^{(b)}, f(x)) = \nabla_w f(w^{(b)}, x)$ for all $x \in \mathbb{R}^D$ and $T \in \mathcal{T}$.*

Theorem 4.2 contrasts optimization with and without neural networks. In standard gradient descent (*eq.* (14)), the map $f$ remains invariant if it is initially invariant. In contrast, with the neural kernel in *eq.* (15), even if $f$ starts invariant, an additional condition on $\nabla_w f$ is needed to maintain invariance. Since this condition is not guaranteed be satisfied throughout the iterations, the optimization can cause $f$ to lose invariance, resulting in different dynamics compared to standard gradient descent.

Many other works have shown that the neural kernel imparts significant changes on the dynamics of gradient descent. For example, Xu et al. (2019a;b) established the frequency principle, showing that the training dynamics of neural networks are significantly biased towards low frequency information, compared to vanilla gradient descent.

## 4.2 STUDYING A CLUSTERED DATASET

In this section, we explore how the neural network kernel $K_{ij}$ in eq. (12) influences the gradient flow on the contrastive learning loss. For clarity, we use a simplified setting that is straightforward enough to provide insights into the neural network's impact on the optimization process. Although simplified, the setting can be easily generalized to extend these insights to broader contexts.

**Dataset Description** Consider a data distribution $\mu \in \mathbb{P}(\mathcal{M})$, where $\mathcal{M}$ is a $d$-dimensional compact submanifold in $\mathbb{R}^D$, and a noisy data distribution $\widetilde{\mu} \in \mathbb{P}(\mathbb{R}^D)$ defined by

$$\widetilde{\mu} = \mu + \alpha,$$

where $\alpha \in \mathbb{P}(\mathcal{M}^\perp)$ represents noise (or perturbation) applied to $\mu$ in the orthogonal direction to $\mathcal{M}$.

For simplicity, assume that

$$\mathcal{M} \subset \left\{ x = (x^1, \cdots, x^d, 0, \cdots, 0) \in \mathbb{R}^D \right\}, \quad \mathcal{M}^\perp \subset \left\{ x = (0, \cdots, 0, x^{d+1}, \cdots, x^D) \in \mathbb{R}^D \right\}.$$

Now, let's impose a clustered structure on the dataset. Given a dataset $\{x_i\}_{i=1}^n$ with $n$ samples i.i.d. from the noisy distribution $\widetilde{\mu}$, we assume that the data is organized into $N$ ($N \leq d$) clusters. Let $\{\xi_q\}_{q=1}^N$ be $N$ points in $\mathcal{M}$ such that $\xi_q \cdot \xi_{q'} \neq 0$ if $q = q'$, and $0$ otherwise. Suppose the data samples are arranged such that for each $i \in \{1, \ldots, n\}$,

$$x_i = \xi_q + \epsilon_i \quad \text{for } n_{q-1} + 1 \leq i \leq n_q \tag{16}$$

where $0 = n_0 < n_1 < \cdots < n_N = n$ and $\epsilon_i$ is a random variable (i.e., noise) and $\|\epsilon_i\| < \delta$ for some positive constant $\delta > 0$. This setup ensures that each data point $x_i$ lies within a ball of radius $\delta$ centered at one of the points $\xi_q$, effectively representing the dataset as $N$ clusters.

**Embedding Map Description** Let $f : \mathbb{R}^{m+D} \to \mathbb{R}^d$ be an embedding map parameterized by a neural network, such that $f = f(w(t), x)$, where $w : \mathbb{R} \to \mathbb{R}^m$ is a vector of neural network parameters. We assume that at $t = 0$, $f$ satisfies

$$f(w(0), x) = R(x^1, \ldots, x^d, 0, \ldots, 0) \in \mathbb{R}^d, \tag{17}$$

for all $x = (x^1, \ldots, x^D) \in \mathbb{R}^D$, where $R : \mathbb{R}^D \to \mathbb{R}^d$ is an arbitrary map. Consequently, this embedding map $f$ is invariant under the following perturbations: for $x \sim \mu$ and $\epsilon \sim \alpha$,

$$f(w(0), x + \epsilon) = R(x) = f(w(0), x).$$

Thus, $f$ is an invariant to the perturbation from $\alpha$. This serves to initialize the embedding map $f$ to be an invariant map that is unrelated to the data distribution $\mu$. This is in some sense the "worst case" initialization, where no information from the data distribution $\mu$ has been imbued upon the latent distribution. The goal is to examine what happens when using this as the initialization for training. As we will see below, the neural kernel always injects information from $\mu$ into the optimization procedure, and can even help recover from poor initializations.

### 4.2.1 PROPERTIES OF THE EMBEDDING MAP

In this section, we derive the explicit formulations for the gradients and the kernel matrix defined in eq. (12), within the setting described in Section 4.2. We examine the training dynamics of eq. (10) to understand how they are influenced by the neural network kernel matrix $K_{ij}$ and the dataset's clustering structure. Specifically, we consider the embedding map $f : \mathbb{R}^{MDd+D} \to \mathbb{R}^d$ parameterized by a one-hidden-layer fully connected neural network:

$$f(w(t), x) = f(B(t), x) = A^\top \sigma(B(t)x), \tag{18}$$

where $A \in \mathbb{R}^{Md \times d}$ is a constant matrix defined as

$$A = \frac{1}{\sqrt{M}} \begin{bmatrix} \mathbb{1}_{M \times 1} & \mathbb{0}_{M \times 1} & \cdots & \mathbb{0}_{M \times 1} \\ \mathbb{0}_{M \times 1} & \mathbb{1}_{M \times 1} & \cdots & \mathbb{0}_{M \times 1} \\ \vdots & \vdots & \ddots & \vdots \\ \mathbb{0}_{M \times 1} & \mathbb{0}_{M \times 1} & \cdots & \mathbb{1}_{M \times 1} \end{bmatrix} \in \mathbb{R}^{Md \times d}, \tag{19}$$

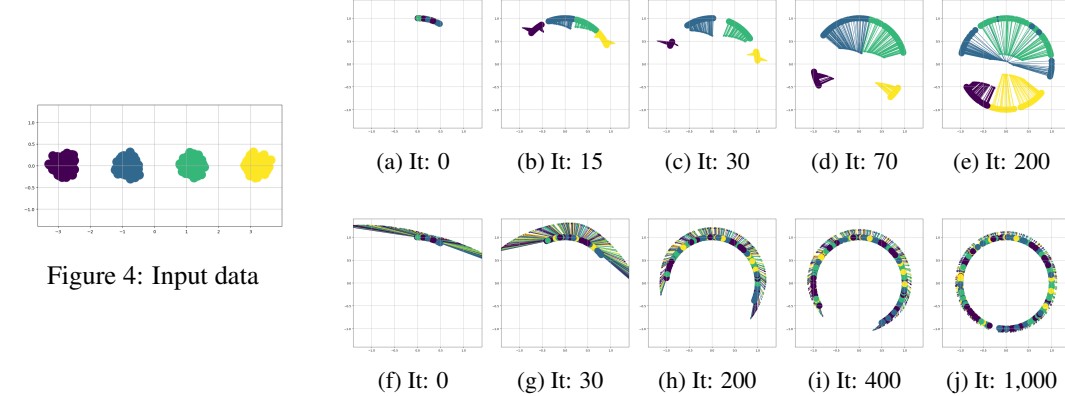

(a) It: 0      (b) It: 15      (c) It: 30      (d) It: 70      (e) It: 200

Figure 4: Input data

(f) It: 0      (g) It: 30      (h) It: 200      (i) It: 400      (j) It: 1,000

Figure 5: Comparison of the optimization process with and without neural network training. The data distribution is shown in (a), with each point colored by its cluster and arrows representing the negative gradient. Row 2 shows the optimization with neural network training, revealing the clustering structure over iterations. Row 3 shows optimization using vanilla gradient descent, where the distribution eventually becomes uniformly dispersed, ignoring the input data's clustering.

where $\mathbb{1}_{M \times 1}$ and $\mathbb{0}_{M \times 1}$ represent the $M$-dimensional vectors of ones and zeros, respectively. Additionally, $B(t) = (b_p^k(t))_{k \in [\![D]\!], p \in [\![Md]\!]} \in \mathbb{R}^{Md \times D}$ is the weight matrix, and $\sigma$ is a differentiable activation function applied element-wise.

Note that $A$ acts as an averaging matrix that, when multiplied by the $(Md)$-dimensional vector $\sigma(B(t)x)$, produces a $d$-dimensional vector. Furthermore, we assume that the parameters of $W$ are uniformly bounded, such that there exists a constant $C$ with $|b_p^k(t)| < C$ for all $t \geq 0$, $k$, and $p$.

*Remark* 4.3. In many contrastive learning studies, the neural network is trained, but the last layer is discarded when retrieving feature representations. Research Bordes et al. (2023); Gui et al. (2023); Wen & Li (2022) shows that discarding the last layer can improve feature quality. While we do not consider this in our analysis for simplicity, exploring its impact on training dynamics is an interesting direction for future work.

Based on the definitions of kernel matrices in eq. (12) and the neural network function $f$ in eq. (18), the following proposition provides the explicit kernel formula.

**Proposition 4.3.** *Given the description of the embedding map in eq. (18), the kernel matrix $K_{ij}^{kl}$ defined in eq. (12) can be explicitly written as*

$$K_{ij}^{kl} = \frac{\mathbb{1}_{k=l}}{M} x_i^\top x_j \sum_{p=(k-1)M+1}^{kM} \sigma'(b_p x_i)\sigma'(b_p x_j). \tag{20}$$

*where $\mathbb{1}_{k=l}$ is an indicator function that equals 1 if $k = l$ and 0 otherwise.*

From Proposition 4.3, as done in NTK paper (Jacot et al., 2018), one can consider how the kernel converges as the width of the neural network approaches infinity, i.e., as $M \to \infty$ in eq. (18). The following proposition shows the formulation of the limiting kernel in the infinite-width neural network.

**Proposition 4.4.** *Suppose the weight matrix $B$ satisfies that each row vector $b_i$, for $i \in \{1, \ldots, Md\}$, consists of independent and identically distributed random variables in $\mathbb{R}^D$ with a Gaussian distribution. Also, suppose the activation function is $\sigma(x) = x_+ = \max\{x, 0\}$. Then, as $M \to \infty$, the kernel matrix converges to $K^\infty \in \mathbb{R}^{d \times d}$, where*

$$K_{ij}^\infty = (x_i^\top x_j) \left[ \frac{1}{2} - \frac{1}{2\pi} \arccos\left( \frac{x_i^\top x_j}{\|x_i\|\|x_j\|} \right) \right] \boldsymbol{I}_{d \times d}, \quad \boldsymbol{I}_{d \times d} \text{ is an identity matrix.}$$

Using the kernel matrix defined in Proposition 4.3, the following theorem presents the explicit form of the gradient flows in terms of the clustering structure and the neural network parameters.

**Theorem 4.5.** *Let the dataset to be clustered be as described in eq. (16). Define a function $\gamma : [\![n]\!] \to [\![N]\!]$ such that $\gamma(i) = q$ if $x_i$ is from the cluster point $\xi_q$. Then, the gradient flow formulation takes the form*

$$\dot{z}_i(t) = -\left(\frac{n_{\gamma(i)} - n_{\gamma(i)-1}}{n}\right) \|\xi_{\gamma(i)}\|^2 \boldsymbol{\beta}_{\gamma(i)} \nabla L(w(t), \xi_{\gamma(i)}) + O(\delta) \qquad (21)$$

*where $\boldsymbol{\beta}_q \in \mathbb{R}^{d \times d}$ ($q \in [\![n]\!]$) is a diagonal matrix where each diagonal entry $\beta_i^k \in \mathbb{R}$ ($k \in [\![d]\!]$) is defined as $\beta_q^k = \frac{1}{M} \sum_{p=(k-1)M+1}^{kM} \sigma'(b_p \xi_q)^2$.*

Similar to Proposition 4.4, one can consider the gradient flow formulation in the limit of infinite width, i.e., as $M \to \infty$. The following corollary provides the explicit form of the neural network gradient flow in the infinite width case.

**Corollary 4.6.** *Under the same conditions as in Proposition 4.4 and Theorem 4.5, if the width approaches infinity, i.e., $M \to \infty$, then the gradient formulation in eq. (21) becomes*

$$\dot{z}_i(t) = -\left(\frac{n_{\gamma(i)} - n_{\gamma(i)-1}}{2n}\right) \|\xi_{\gamma(i)}\|^2 \nabla L(f(w(t), \xi_{\gamma(i)})) + O(\delta).$$

*Remark* 4.4. Theorem 4.5 and Proposition 4.4 show that the gradient flow in the clustering setting is scaled by the cluster size ratio, $(n_{\gamma(i)} - n_{\gamma(i)-1})/n$. This implies that clusters with fewer points contribute less to the gradient, potentially failing to capture smaller clusters effectively. This aligns with Assran et al. (2022), which shows that semi-supervised methods, including contrastive learning, perform worse with imbalanced class distributions and better with uniform ones.

The gradient flow formulations in Theorem 4.5 and Corollary 4.6 modify the vanilla gradient flow in eq. (13), guiding the iterations toward a stationary solution that is both invariant to perturbations *and* influenced by the dataset's geometry. Specifically, the first term in eq. (21) shows that for points in the same cluster, the gradient is the same, with an error of order $O(\delta)$. If the neural network is initialized randomly, the gradient at points from the same cluster will align, leading to embeddings that reflect the data's clustering structure.

This behavior is shown in Figure 5, where a 2D dataset with four clusters along the $x$-axis at $-3$ (purple), $-1$ (blue), $1$ (green), and $3$ (yellow) is analyzed. The arrow at each point represents the negative gradient. The number of iterations differs between training dynamics (with and without neural network optimization) due to the neural network kernel's impact on convergence speed. The iterations are chosen to best capture the evolution of the feature distribution, with both figures stabilizing after 200 iterations (with neural network) and 1,000 iterations (without).

The training dynamics show that, despite random initialization, data points from the same cluster follow a similar gradient, driving separation of different clusters early on. By iteration 15, the clustering structure is preserved in the embedded distribution. In contrast, without neural network optimization, the data points spread out and form a uniform distribution on a sphere, ignoring the input data's cluster structure. This aligns with Theorem 3.2, which indicates that the gradient of the loss function is independent of the input structure.

## 5 CONCLUSION AND FUTURE WORK

We have studied the SimCLR contrastive learning problem from the perspective of a variational analysis and through the dynamics of training a neural network to represent the embedding function. Our findings strongly suggest that in order to fully understand the representation power of contrastive learning, it is necessary to study the training dynamics of gradient descent, as vanilla gradient descent *forgets* all information about the data distribution.

The results in this paper are preliminary, and there are many interesting directions for future work. It would be natural to examine a mean field limit (Mei et al., 2018) for the training dynamics, which may shed more light on this phenomenon (e.g., in theorem 4.5). We can also consider an infinite width neural network, as is done in the NtK setting, in theorem 4.5 to attempt to rigorously establish convergence of the training dynamics. It would also be interesting to explore applications of these techniques to other deep learning methods.

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

# A APPENDIX

In the appendix, we present the proofs of that are missing in the main manuscript.

## A.1 INTERPRETATION OF VICREG AND BYOL

In this section, we show that two other popular methods related to deep learning models for learning dataset invariance exhibit a similar phenomenon as shown in Proposition 2.1, namely that the loss function itself is ill-posed. First, consider the VICReg Bardes et al. (2021) loss. Given a data distribution $\mu \in \mathbb{P}(\mathbb{R}^D)$ and a distribution for the perturbation functions $\nu$, VICReg minimizes

$$
\min_{f:\mathbb{R}^D \to \mathbb{R}^d} \mathbb{E}_{f,g\sim\nu} \mathbb{E}_{x_1,\cdots,x_n\sim\mu} \frac{\lambda_1}{n} \sum_{i=1}^{n} \|f(f(x_i)) - f(g(x_i))\|^2
$$
$$
+ \lambda_2\Big(v(f(f(x_1)),\cdots,f(f(x_n))) + v(f(g(x_1)),\cdots,f(g(x_n)))\Big)
$$
$$
+ \lambda_3\Big(c(f(f(x_1)),\cdots,f(f(x_n))) + c(f(g(x_1)),\cdots,f(g(x_n)))\Big)
$$

where $\lambda_1$, $\lambda_2$, and $\lambda_3$ are hyperparameters. The first term ensures the invariance of $f$ with respect to perturbation functions from $\nu$, $v$ maintains the variance of each embedding dimension, and $c$ regularizes the covariance between pairs of embedded points towards zero. Suppose $f$ is an invariant embedding map such that $f(T(x)) = f(x)$ for all $T \sim \nu$. Then, the above minimization problem becomes

$$
\min_{f:\mathbb{R}^D \to \mathbb{R}^d} \mathbb{E}_{f,g\sim\nu} \mathbb{E}_{x_1,\cdots,x_n\sim\mu} \lambda_2\Big(v(f(x_1),\cdots,f(x_n)) + v(f(x_1),\cdots,f(x_n))\Big)
$$
$$
+ \lambda_3\Big(c(f(x_1),\cdots,f(x_n)) + c(f(x_1),\cdots,f(x_n))\Big)
$$
$$
= \min_{f:\mathbb{R}^D \to \mathbb{R}^d} \mathbb{E}_{y_1,\cdots,y_n\sim f_\#\mu} \lambda_2\Big(v(y_1,\cdots,y_n) + v(y_1,\cdots,y_n)\Big)
$$
$$
+ \lambda_3\Big(c(y_1,\cdots,y_n) + c(y_1,\cdots,y_n)\Big)
$$

Similar to the result in Proposition 2.1, the invariance term vanishes. This minimization can now be expressed as a minimization over the embedded distribution:

$$
\min_{\rho\in\mathbb{P}(\mathbb{R}^d)} \mathbb{E}_{y_1,\cdots,y_n\sim\rho} \lambda_2\Big(v(y_1,\cdots,y_n) + v(y_1,\cdots,y_n)\Big)
$$
$$
+ \lambda_3\Big(c(y_1,\cdots,y_n) + c(y_1,\cdots,y_n)\Big)
$$

This shows that given an invariant map $f$, the minimization problem becomes completely independent of the input data $\mu$, thus demonstrating the same ill-posedness as the NT-Xent loss in Proposition 2.1.

Now, consider the loss function from BYOL Grill et al. (2020). Given a data distribution $\mu \in \mathbb{P}(\mathbb{R}^D)$ and a distribution for the perturbation functions $\nu$, the loss takes the form

$$
\min_{f,q} \mathbb{E}_{f,g\sim\nu} \mathbb{E}_{x\sim\mu} \|q(f(T(x))) - f(T'(x))\|^2
$$

where $q : \mathbb{R}^d \to \mathbb{R}^d$ is an auxiliary function designed to prevent $f$ from collapsing all points $x$ to a constant in $\mathbb{R}^d$. Similar to the previous case, if we assume an invariant map $f$, the above problem becomes

$$\min_{f,q} \mathbb{E}_{x \sim \mu} \|q(f(x)) - f(x)\|^2 = \min_{f,q} \mathbb{E}_{y \sim f_{\#}\mu} \|q(y) - y\|^2$$

where the second equality follows from a change of variables. Again, this minimization problem can be written with respect to the embedded distribution as:

$$\min_{\rho \in \mathbb{P}(\mathbb{R}^d), q} \mathbb{E}_{y \sim \rho} \|q(y) - y\|^2$$

This again shows that once the invariant map is considered, the minimization problem becomes completely independent of the input data $\mu$, highlighting the ill-posedness of the cost function.

## A.2 FURTHER ANALYSIS OF THE STATIONARY POINTS OF EQUATION (4)

From the modified formulation eq. (4), we can define a minimizer that minimizes the function $L(f)$ on a constraint set $\mathcal{C} = \{f : \mathbb{R}^D \to \mathbb{R}^d\}$. The following proposition provides insight into the minimizer of eq. (4). The proof is provided in the appendix.

**Proposition A.1.** *This proposition describes three different possible local minimizers of eq. (4) that satisfy the Euler-Lagrange equation in eq. (6).*

1. *Any map $f : \mathbb{R}^D \to \mathbb{R}^d$ that maps to a constant, such that*

$$f(x) = c \in \mathbb{R}^d, \quad \forall x \in \mathcal{M}.$$

2. *In addition to the condition in eq. (5), suppose the attraction and repulsion similarity functions $a : \mathbb{R}_{\geq 0} \to \mathbb{R}$ and $r : \mathbb{R}_{\geq 0} \to \mathbb{R}$ satisfy the following properties:*

   (a) *Each function is maximized at 0, where its value is 1.*
   (b) *Each function satisfies $\lim_{t \to \infty} a(t) = 0$ and $\lim_{t \to \infty} t a'(t) = 0$.*

   *Let $f$ be a map invariant to $\mathcal{T}$. Consider a sequence of maps $\{f_k\}$ such that*

$$f_k(x) = k f(x), \quad \forall x \in \mathcal{M}, \forall k \in \mathbb{N}.$$

   *The limit $f_* = \lim_{k \to \infty} f_k$ satisfies the Euler-Lagrange equation eq. (6).*

*Proof of Proposition A.1.* If $f$ is a constant function, it is trivial that it satisfies eq. (6).

Let us prove the second part of the proposition. From the Euler-Lagrange equation in eq. (6), by plugging in $f_k$ and using the fact that $f$ is invariant to $\mathcal{T}$, the Euler-Lagrange equation can be simplified to

$$\int_{\mathbb{R}^D} \left( \Psi'(G(f_k, \mathrm{Id}, x)) + \Psi'(G(f_k, \mathrm{Id}, y)) \right) r'_{f_k}(x, y) \langle f_k(x) - f_k(y), h(x) \rangle \, d\mu(y)$$

for any $h : \mathbb{R}^D \to \mathbb{R}^d$. Using the invariance of $f_k$, we have

$$= \int_{\mathbb{R}^D} \left( \Psi'(G(f_k, \mathrm{Id}, x)) + \Psi'(G(f_k, \mathrm{Id}, y)) \right) \left( k r'_{f_k}(x, y) \right) \langle f(x) - f(y), h(x) \rangle \, d\mu(y). \quad (22)$$

Furthermore, by the assumptions on the function $r$,

$$k r'_{f_k}(x, y) = k r' \left( \frac{k^2 \|f(x) - f(y)\|^2}{2} \right) \to 0, \quad \text{as } k \to \infty$$

$$\Psi'(G(f_k, \mathrm{Id}, x)) = \Psi' \left( \mathbb{E}_{z \sim \mu} r \left( \frac{k^2 \|f(x) - f(z)\|^2}{2} \right) \right) \to \Psi'(0), \quad \text{as } k \to \infty.$$

Thus, eq. (22) converges to 0 as $k \to \infty$. This proves the theorem. $\qquad \square$

## A.3 Proof of Theorem 3.2

First, we prove Theorem 3.2, which characterizes the stationary points of the loss function. After the proof, we demonstrate that by considering an additional condition on the direction of the second variation at the stationary points, it is the second variation is strictly positive, thereby showing that the stationary point is a local minimizer under this condition.

*Proof of Theorem 3.2.* We consider the following problem:

$$\min_{f:\mathbb{R}^D \to \mathbb{S}^{d-1}} L(f), \tag{23}$$

where $L$ is a loss function defined in eq. (4). The problem in eq. (23) can be reformulated as a constrained minimization problem:

$$\min_{\substack{f:\mathbb{R}^D \to \mathbb{R}^d \\ \|f\|=1}} L(f).$$

By relaxing the the constraint for $\|f\| = 1$, we can derive the lower bound such that

$$\min_{\substack{f:\mathbb{R}^D \to \mathbb{R}^d \\ \|f\|=1}} L(f) \geq \min_{\substack{f:\mathbb{R}^D \to \mathbb{R}^d \\ \int_{\mathbb{R}^D} \|f\|d\mu=1}} L(f).$$

Note that since the constraint sets satisfy $\{\|f\| = 1\} \subset \{\int_{\mathbb{R}^D} \|f\|d\mu = 1\}$, the stationary point from from the latter constraint set is also the stationary point of the prior set.

By introducing the Lagrange multiplier $\lambda$ for the constraint, we can convert the minimization problem into a minimax problem:

$$\min_{f:\mathbb{R}^D \to \mathbb{R}^d} \max_{\lambda \in \mathbb{R}} \left[ L(f) + \lambda \mathbb{E}_{x \sim \mu}(1 - \|f(x)\|) \right]. \tag{24}$$

Using the Euler-Lagrange formulation in eq. (6), we can derive the Euler-Lagrange equation for the above problem, incorporating the Lagrange multiplier. To show that $f$ is a minimizer of the problem in eq. (24), we need to demonstrate that there exists $\lambda \in \mathbb{R}$ such that the following equation holds:

$$\int_{\mathbb{R}^D} \left[ \Psi'(G(f,x)) + \Psi'(G(f,y)) \right] \eta'_f(x,y)(f(x) - f(y)) \, d\mu(y) - \lambda \frac{f(x)}{\|f(x)\|} = 0,$$

for all $x \in \mathcal{M}$. Note that since $f$ is an invariant map, $f$ disappears and $\eta_f(x, f(x)) = 1$. Furthermore, since $f$ maps onto $\mathbb{S}^{d-1}$, we have $\|f(x)\| = 1$ for all $x \in \mathbb{R}^D$. Additionally, using the change of variables, we obtain

$$\lambda = C \int_{\mathbb{S}^{d-1}} r'(|x-y|^2/2)(x-y) \, df_{\#}\mu(y), \tag{25}$$

where $C$ is defined as $C = \Psi'(\mathbb{E}_{z \sim f_{\#}\mu} \left[ r(|x_0 - z|^2/2) \right])$ for $x_0 \sim f_{\#}\mu$. Given that the function

$$h(x,y) = r'(|x-y|^2/2)(x-y)$$

is an anti-symmetric function, by the assumption on $f_{\#}\mu$ in eq. (8), the integral on the right-hand side of eq. (25) is constant for all $x \sim f_{\#}\mu$. Therefore, by defining $\lambda$ as in eq. (25), this proves the lemma. □

Now that we have identified the characteristics required for embedding maps to be stationary points, the next lemma shows that the second variation at this stationary point, in a specific direction $h$, is positive. This demonstrates that the stationary point is indeed a local minimizer along this particular direction.

**Lemma A.2.** *Fix $\tau > 0$ and define $\eta_f(x,y) = e^{-\|f(x)-f(y)\|^2/2\tau}$. Let $f : \mathbb{R}^D \to \mathbb{S}^{d-1}$ be an embedding map such that the embedded distribution $f_{\#}\mu = \sum_{i=1}^n \delta_{x_i}$ is a discrete measure on $\mathbb{S}^{d-1}$, satisfying that the number of points $n = Km$, where $K$ is the number of cluster centers $\{\xi_i\}_{i=1}^K$ and $m$ is some positive integer. Moreover, the points satisfy the condition:*

$$x_i = \xi_{\lfloor i/K \rfloor + 1} \quad for \ i \in [\![n]\!]. \tag{26}$$

*Furthermore, let $\sigma > 0$ be a positive constant satisfying $\sigma > 3K^2\tau$. Then,*

$$\delta^2 L(f)(h, h) > 0$$

*for any $h : \mathbb{R}^D \to \mathbb{R}^d$ satisfying $f + h \in \mathbb{S}^{d-1}$ and*

$$\left( \langle f(\xi_i) - f(\xi_j), h(\xi_i) - h(\xi_j) \rangle \right)^2 \geq \sigma \|h(\xi_i) - h(\xi_j)\|^2. \tag{27}$$

*Proof.* Let $f : \mathbb{R}^D \to \mathbb{R}^d$ be an invariant embedding map. From the proof of **??**, the first variation takes the form

$$\int_{\mathbb{R}^D} \Psi'(G(f, x)) \int_{\mathbb{R}^D} \eta'_f(x, y) \langle f(x) - f(y), h(x) - h(y) \rangle d\mu(y) d\mu(x).$$

The second variation takes the form

$$\int_{\mathbb{R}^D} \Psi''(G(f, x)) \left( \int_{\mathbb{R}^D} \eta'_f(x, y) \langle f(x) - f(y), h(x) - h(y) \rangle d\mu(y) \right)^2 d\mu(x)$$

$$+ \int_{\mathbb{R}^D} \Psi'(G(f, x)) \int_{\mathbb{R}^D} r''_f(x, y) \Big( \langle f(x) - f(y), h(x) - h(y) \rangle \Big)^2 d\mu(y) d\mu(x)$$

$$+ \int_{\mathbb{R}^D} \Psi'(G(f, x)) \int_{\mathbb{R}^D} \eta'_f(x, y) \|h(x) - h(y)\|^2 d\mu(y) d\mu(x).$$

For simplicity, let us choose explicit forms for $\Psi$ and $r$. The proof will be general enough to apply to any $\Psi$ and $r$ that satisfy the conditions mentioned in the paper. Let $\Psi(t) = \log(1 + t/2)$ and $r(t) = e^{-t/(2\tau)}$. With these choice of functions and by the change of variables,

$$= -\frac{1}{\tau^2} \int_{\mathbb{S}^{d-1}} \left( \frac{1}{1 + G(x)^2/2} \right)^2 \left( \int_{\mathbb{S}^{d-1}} e^{-\|x-y\|^2/(2\tau)} \langle x - y, T'(x) - T'(y) \rangle df_{\#}\mu(y) \right)^2 df_{\#}\mu(x)$$

$$+ \frac{1}{\tau^2} \int_{\mathbb{S}^{d-1}} \frac{1}{1 + G(x)^2/2} \int_{\mathbb{S}^{d-1}} e^{-\|x-y\|^2/(2\tau)} \Big( \langle x - y, T'(x) - T'(y) \rangle \Big)^2 df_{\#}\mu(y) df_{\#}\mu(x)$$

$$- \frac{1}{\tau} \int_{\mathbb{S}^{d-1}} \frac{1}{1 + G(x)^2/2} \int_{\mathbb{S}^{d-1}} e^{-\|x-y\|^2/(2\tau)} \|T'(x) - T'(y)\|^2 df_{\#}\mu(y) df_{\#}\mu(x). \tag{28}$$

where $G(x) = \mathbb{E}_{y \sim f_{\#}\mu} e^{-\|x-y\|^2/(2\tau)}$ and $T'(x) = h(f^{-1}(x))$. By Jensen's inequality, we have

$$\left( \int_{\mathbb{S}^{d-1}} e^{-\|x-y\|^2/(2\tau)} \langle x - y, T'(x) - T'(y) \rangle df_{\#}\mu(y) \right)^2$$

$$\leq \int_{\mathbb{S}^{d-1}} e^{-\|x-y\|^2/(2\tau)} \Big( \langle x - y, T'(x) - T'(y) \rangle \Big)^2 df_{\#}\mu(y).$$

Therefore, eq. (28) can be bounded below by

$$\geq \frac{1}{\tau^2} \int_{\mathbb{S}^{d-1}} \frac{G(x)^2/2}{(1 + G(x)^2/2)^2} \int_{\mathbb{S}^{d-1}} e^{-\|x-y\|^2/(2\tau)} \Big( \langle x - y, T'(x) - T'(y) \rangle \Big)^2 df_{\#}\mu(y) df_{\#}\mu(x)$$

$$- \frac{1}{\tau} \int_{\mathbb{S}^{d-1}} \frac{1}{1 + G(x)^2/2} \int_{\mathbb{S}^{d-1}} e^{-\|x-y\|^2/(2\tau)} \|T'(x) - T'(y)\|^2 df_{\#}\mu(y) df_{\#}\mu(x)$$

$$= \frac{1}{\tau^2} \int_{\mathbb{S}^{d-1}} \frac{1}{1 + G(x)^2/2} \int_{\mathbb{S}^{d-1}} e^{-\|x-y\|^2/(2\tau)}$$

$$\left( \frac{G(x)^2}{2(1 + G(x)^2/2)} \Big( \langle x - y, T'(x) - T'(y) \rangle \Big)^2 - \tau \|T'(x) - T'(y)\|^2 \right) df_{\#}\mu(y) df_{\#}\mu(x).$$

By the assumption on $f_{\#}\mu$ in eq. (26), the above can be written as

$$
=\frac{1}{n^2\tau^2}\sum_{i=1}^{n}\frac{1}{1+\widetilde{G}(x_i)^2/2}\sum_{\substack{j=1\\j\neq i}}^{n}e^{-\|x_i-x_j\|^2/(2\tau)}
$$

$$
\left(\frac{\widetilde{G}(x_i)^2}{2(1+G(x)^2/2)}\Big(\langle x_i-x_j,g(x_i)-g(x_j)\rangle\Big)^2-\tau\|g(x_i)-g(x_j)\|^2\right)
$$

$$
=\frac{m^2}{n^2\tau^2}\sum_{i=1}^{K}\frac{1}{1+\widetilde{G}(\xi_i)^2/2}\sum_{\substack{j=1\\j\neq i}}^{K}e^{-\|\xi_i-\xi_j\|^2/(2\tau)}
$$

$$
\left(\frac{\widetilde{G}(\xi_i)^2}{2(1+\widetilde{G}(\xi_i)^2/2)}\Big(\langle \xi_i-\xi_j,g(\xi_i)-g(\xi_j)\rangle\Big)^2-\tau\|g(\xi_i)-g(\xi_j)\|^2\right)
$$

(29)

If $K=1$, the second variation becomes 0, and is therefore nonnegative. Now, suppose $K>1$. We can bound $\widetilde{G}$ from below by

$$
\widetilde{G}(\xi_i)=\frac{1}{K}\sum_{k=1}^{K}e^{-\|\xi_i-\xi_k\|^2/(2\tau)}\geq\frac{1}{K}.
$$

(30)

Furthermore, from the condition in eq. (27), we have

$$
\Big(\langle \xi_i-\xi_j,g(\xi_i)-g(\xi_j)\rangle\Big)^2\geq\sigma\|g(\xi_i)-g(\xi_j)\|^2
$$

(31)

for some positive constant $\sigma>0$. Combining eq. (30) and eq. (31), we can bound eq. (29) from below by

$$
\geq\frac{1}{K^2\tau^2}\sum_{i=1}^{K}\frac{1}{1+\widetilde{G}(\xi_i)^2/2}\sum_{\substack{j=1\\j\neq i}}^{K}e^{-\|\xi_i-\xi_j\|^2/(2\tau)}\left(\frac{\sigma}{3K^2}-\tau\right)\|g(\xi_i)-g(\xi_j)\|^2.
$$

By the condition on $\sigma$, the above quantity is strictly greater than zero. This concludes the proof of the lemma.

$\square$

## A.4 PROOF OF PROPOSITION 4.1

*Proof.* The gradient of $\mathcal{L}$ is given by

$$
\nabla\mathcal{L}(w)=\frac{1}{n}\sum_{i=1}^{n}\sum_{k=1}^{d}\nabla_{y^k}L(f(w,x_i))\nabla_w f^k(w,x_i)
$$

(32)

where $\nabla_{y^k}L(f(w,x_i))$ is a gradient of $L$ with respect to $y_k$ coordinate. For simplicity of notation, let us denote by

$$
f_i^k=f^k(w,x_i),\quad L_i=L(f(w,x_i)).
$$

Thus, eq. (32) can be rewritten as

$$
\nabla\mathcal{L}(w)=\frac{1}{n}\sum_{i=1}^{n}\sum_{k=1}^{d}\nabla_{y^k}L_i\nabla_w f_i^k
$$

(33)

By the definition of the loss function in eq. (9), $w(t)$ satisfies the gradient flow such that

$$
\dot{w}(t)=-\nabla\mathcal{L}(w).
$$

(34)

Thus, the solution of the above ODE converges to the local minimizer of $\mathcal{L}$ as $t$ grows.

For each $i \in [\![n]\!]$ and $k \in [\![d]\!]$, denote by

$$z_i^k(t) = f_i^k(t).$$

Let us compute the time derivative of $z_i^k$. Using a chain rule, eq. (33) and eq. (34),

$$\dot{z}_i^k(t) = \nabla_w f_i^k \cdot \dot{w}(t) = -\nabla_w f_i^k \cdot \nabla \mathcal{L}(w) = -\frac{1}{n} \sum_{j=1}^{m} \sum_{l=1}^{d} \nabla_w f_i^k \cdot \nabla_w f_j^l \nabla_{y^l} L_j. \qquad (35)$$

Using eq. (12), eq. (35), $\dot{z}_i(t)$ can be written as

$$\dot{z}_i(t) = -\frac{1}{n} \sum_{j=1}^{n} K_{ij} \nabla L_j(t).$$

This completes the proof. $\qquad \square$

## A.5 PROOF OF THEOREM 4.2

*Proof.* Consider the gradient descent iterations in eq. (14). Suppose $f$ is invariant to the perturbation from $\nu$ at $b$-th iteration, that is we have $f(w^{(b)}, f(x)) = f(w^{(b)}, x)$ for all $x \sim \mu$ and $T \sim \nu$. We want to show that, given an invariant embedding map $f(w^{(b)}, \cdot)$, it remains invariant after iteration $b$. From the gradient formulation of the loss function in Proposition 3.1, we have

$$\nabla L(f(w^{(b)}, x)) = -\int_{\mathbb{R}^D} \Psi'(x, y)(f(w^{(b)}, x) - f(w^{(b)}, y)) \, d\mu(y),$$

where $\Psi'(x, y) = \Big( \Psi'(G(f(w^{(b)}, \cdot), x)) + \Psi'(G(f(w^{(b)}, \cdot), y)) \Big) \eta'_{f(w^{(b)}, \cdot)}(x, y)$.

From the gradient descent formulation in eq. (14), we have

$$f(w^{(b+1)}, f(x_i)) = f(w^{(b)}, f(x_i)) - \sigma \nabla L(f(w^{(b)}, f(x_i))),$$

which gives

$$f(w^{(b+1)}, f(x_i)) = f(w^{(b)}, f(x_i)) + \sigma \int_{\mathbb{R}^D} \Psi'(x, y)(f(w^{(b)}, f(x_i)) - f(w^{(b)}, y)) \, d\mu(y),$$

and since $f(w^{(b)}, f(x_i)) = f(w^{(b)}, x_i)$ by invariance, this simplifies to

$$f(w^{(b+1)}, x_i) - \sigma \nabla L(f(w^{(b)}, x_i)) = f(w^{(b+1)}, x_i),$$

which shows that $f(w^{(b+1)}, x_i)$ is invariant for all $i \in [\![n]\!]$. Therefore, the embedding map remains invariant throughout the optimization process.

Now, consider the gradient descent iteration with a neural network in eq. (15). Suppose $f$ is invariant to perturbations from $\nu$ and satisfies

$$\nabla_w f(w^{(b)}, f(x)) = \nabla_w f(w^{(b)}, x), \quad \forall x \sim \mu, f \sim \nu. \qquad (36)$$

Denote the kernel matrix function $K_{ij}$ given a perturbation function $T \sim \nu$ as

$$(K_{ij}(w^{(b)}, f))^{kl} = (\nabla_w f^k(w^{(b)}, f(x_i)))^\top (\nabla_w f^l(w^{(b)}, f(x_j))).$$

Then, we have

$$f(w^{(b+1)}, f(x_i)) = f(w^{(b)}, f(x_i)) - \frac{\sigma}{n} \sum_{j=1}^{n} K_{ij}(w^{(b)}, f) \nabla L(f(w^{(b)}, f(x_i)))$$

$$= f(w^{(b)}, x_i) - \frac{\sigma}{n} \sum_{j=1}^{n} K_{ij}(w^{(b)}, \text{Id}) \nabla L(f(w^{(b)}, x_i))$$

$$= f(w^{(b+1)}, x_i).$$

Thus, if $f$ is invariant at the $b$-th iteration, it remains invariant. However, note that this result no longer holds if the condition in eq. (36) fails, meaning that $f$ is not invariant for $b + 1$-th iteration. This completes the proof. $\qquad \square$

A.6   PROOF OF PROPOSITION 4.3

*Proof.* We describe the matrices $A \in \mathbb{R}^{Md \times d}$ and $B \in \mathbb{R}^{Md \times D}$ as follows:

$$A = \begin{bmatrix} | & | & \cdots & | \\ a^1 & a^2 & \cdots & a^d \\ | & | & \cdots & | \end{bmatrix} = \begin{bmatrix} - & a_1 & - \\ & \vdots & \\ - & a_{Md} & - \end{bmatrix} = \begin{bmatrix} a_1^1 & \cdots & a_1^d \\ \vdots & \cdots & \vdots \\ a_{Md}^1 & \cdots & a_{Md}^d \end{bmatrix},$$

$$B(t) = \begin{bmatrix} | & | & \cdots & | \\ b^1(t) & b^2(t) & \cdots & b^D(t) \\ | & | & \cdots & | \end{bmatrix} = \begin{bmatrix} - & b_1(t) & - \\ & \vdots & \\ - & b_{Md}(t) & - \end{bmatrix} = \begin{bmatrix} b_1^1(t) & \cdots & b_1^D(t) \\ \vdots & \cdots & \vdots \\ b_{Md}^1(t) & \cdots & b_{Md}^D(t) \end{bmatrix}.$$

In this notation, $a^k$ and $b^k$ are $Md$-dimensional column vectors, $a_p$ and $b_p$ are $d$- and $D$-dimensional row vectors, and $a_p^k$ and $b_p^k$ are scalars.

We can write $f^k$ with respect to $a_i^k$ and $b_i^k$.

$$f^k(B, x) = (a^k)^\top \sigma(Bx) = \sum_{i=1}^{Md} a_i^k \sigma\Big(b_i x\Big) = \frac{1}{\sqrt{M}} \sum_{i=(k-1)M+1}^{kM} \sigma\Big(b_i x\Big)$$

where the last equality uses the definition of a matrix $A$ in eq. (19). By differentiating with respect to $b_i^l$, we can derive explicit forms for the gradient of $f^k$ with respect to a weight matrix $B$.

$$\nabla_w f^k(B, x) = \Big(a^k \odot \sigma'(Bx)\Big) x^\top$$

$$= \begin{bmatrix} a_1^k \sigma'(b_1 x) x^1 & \cdots & a_1^k \sigma'(b_1 x) x^D \\ \vdots & \ddots & \vdots \\ a_M^k \sigma'(b_M x) x^1 & \cdots & a_M^k \sigma'(b_M x) x^D \end{bmatrix}$$

$$= \frac{1}{\sqrt{M}} \begin{bmatrix} 0 & \cdots & 0 \\ \vdots & \ddots & \vdots \\ 0 & \cdots & 0 \\ \sigma'(b_1 x) x^1 & \cdots & \sigma'(b_1 x) x^D \\ \vdots & \ddots & \vdots \\ \sigma'(b_M x) x^1 & \cdots & \sigma'(b_M x) x^D \\ 0 & \cdots & 0 \\ \vdots & \ddots & \vdots \\ 0 & \cdots & 0 \end{bmatrix} \in \mathbb{R}^{Md \times D}$$

where the row index of nonzero entries ranges from $(k-1)M + 1$ to $kM$.

Define an inner product such that for $h \in \mathbb{R}^{Md \times D}$

$$\langle \nabla_w f^k(B, x), h \rangle, \quad k \in [\![D]\!].$$

Now we are ready to show the explicit formula of the inner product $\langle \nabla_w f^k, \nabla_w f^l \rangle$.

$$\langle \nabla_w f^k(B, x_i), \nabla_w f^l(B, x_j) \rangle = \frac{\mathbb{1}_{k=l}}{M} (x_i^\top x_j) \sum_{p=(k-1)M+1}^{kM} \sigma'(b_p x_i) \sigma'(b_p x_j)$$

where $\mathbb{1}_{k=l}$ is an indicator function that equals 1 if $k = l$ and 0 otherwise. Therefore, the kernel matrix takes the form

$$(K^{kl})_{ij} = \frac{\mathbb{1}_{k=l}}{M} (x_i^\top x_j) \sum_{p=(k-1)M+1}^{kM} \sigma'(b_p x_i) \sigma'(b_p x_j).$$

□

*Proof of Theorem 4.5.* Using the definition of a function $\gamma$ in Theorem 4.5 and using the structure of the dataset $\{x_i\}$ in eq. (16), consider $x_i$ and $x_j$ in $\gamma(i)$-th cluster and $\gamma(j)$-th cluster respectively. Since the dataset is sampled from a compactly supported data distribution and given the assumption that the activation function has bounded derivatives, we have

$$x_i^\top x_j = \xi_{\gamma(i)}^\top \xi_{\gamma(j)} + O(\delta) = \mathbb{1}_{\gamma(i)=\gamma(j)} + O(\delta)$$

$$\sigma'(b_p x_i) = \sigma'(b_p \xi_{\gamma(i)}) + O(\delta).$$

Thus,

$$K_{ij}^{kl} = \frac{\mathbb{1}_{k=l}}{M} \xi_{\gamma(i)}^\top \xi_{\gamma(j)} \mathbb{1}_{\gamma(i)=\gamma(j)} \sum_{p=(k-1)M+1}^{kM} \sigma'(b_p \xi_{\gamma(i)})\sigma'(b_p \xi_{\gamma(j)}) + O(\delta).$$

Combining all, we can write the kernel matrix $K_{ij}$ as the following

$$K_{ij} = \mathbb{1}_{\gamma(i)=\gamma(j)} \|\xi_{\gamma(i)}\|^2 \boldsymbol{\beta}_{\gamma(i)} + O(\delta)$$

where $\boldsymbol{\beta}_i \in \mathbb{R}^{d \times d}$ ($i \in [\![n]\!]$) is a diagonal matrix defined as

$$\boldsymbol{\beta}_i = \begin{bmatrix} \beta_i^1 & 0 & \cdots & 0 \\ 0 & \beta_i^2 & \cdots & 0 \\ \vdots & \vdots & \ddots & \vdots \\ 0 & 0 & \cdots & \beta_i^d \end{bmatrix}$$

where $\beta_i^k \in \mathbb{R}$ ($k \in [\![d]\!]$) is defined as

$$\beta_i^k = \frac{1}{M} \sum_{p=(k-1)M+1}^{kM} \sigma'(b_p \xi_{\gamma(i)})^2. \tag{37}$$

Thus, from the gradient flow formulation in eq. (10), we have

$$\dot{z}_i(t) = -\frac{1}{n} \sum_{j=1}^n K_{ij}(t)\nabla L_j(t)$$

$$= -\frac{\|\xi_{\gamma(i)}\|^2}{n} \boldsymbol{\beta}_{\gamma(i)} \sum_{j \in \gamma(i)} \nabla L_j(t) + O(\delta)$$

$$= -\frac{\|\xi_{\gamma(i)}\|^2}{n} (n_{\gamma(i)} - n_{\gamma(i)-1}) \boldsymbol{\beta}_{\gamma(i)} \nabla L(w(t), \xi_{\gamma(i)}) + O(\delta)$$

$$= -\left(\frac{n_{\gamma(i)} - n_{\gamma(i)-1}}{n}\right) \|\xi_{\gamma(i)}\|^2 \boldsymbol{\beta}_{\gamma(i)} \nabla L(w(t), \xi_{\gamma(i)}) + O(\delta).$$

$\square$

## B    EXTRA NUEMRICAL RESULTS

In this section, we provide additional experimental results to validate Theorem 4.5, showing that neural network optimization influences training dynamics. Even when starting with the same random initialization of the embedded distribution, the training dynamics with neural networks are guided toward stationary points where the clustering structure is imposed. In contrast, vanilla gradient descent without neural network optimization is independent of the data structure.

The comparison of optimization processes with and without neural network training in 2D and 3D is shown, with the data distribution presented in (a) and (l). A 4-layer fully connected neural network was used in this experiment, demonstrating that the same behavior is observed even with different neural network architectures. The color of each point corresponds to its respective cluster. Rows 2 and 5 illustrate optimization with neural network training, starting from a random initial embedding and progressively revealing the clustering structure over iterations. Rows 3 and 6 show the optimization process using vanilla gradient descent without neural network training. Over time, the distribution converges to a uniformly dispersed arrangement, disregarding the clustering structure of the input data.

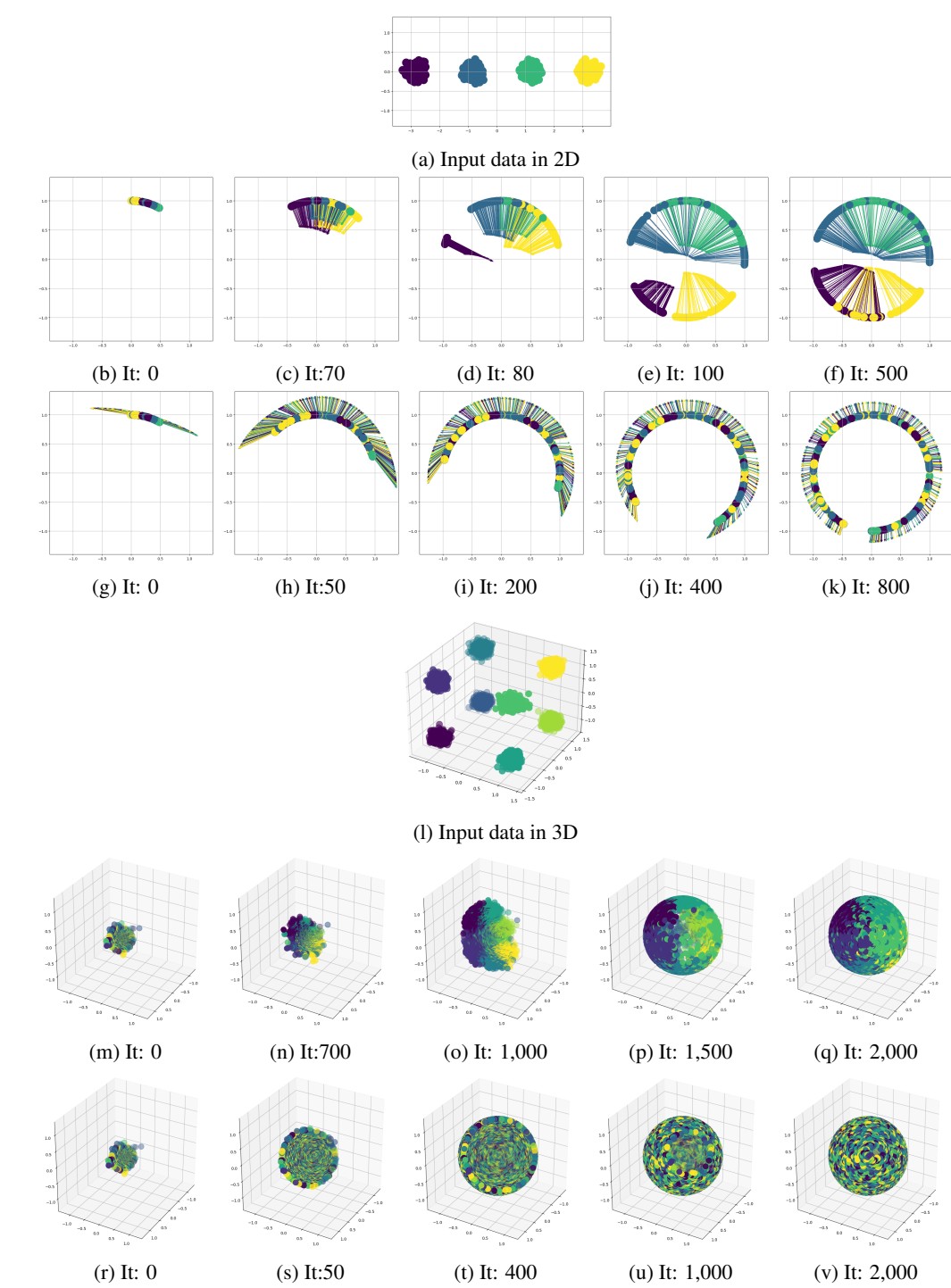

Figure 6: This experiment compares the optimization processes with and without neural network training in 2D and 3D, with the data distribution depicted in (a) and (l). A 4-layer fully connected neural network demonstrates consistent outcome as in Figure 5. Each point's color indicates its cluster. Rows 2 and 5 show optimization with neural network training, starting from a random embedding and gradually revealing the clustering structure. In contrast, Rows 3 and 6 illustrate the optimization process using vanilla gradient descent, which converges to a uniformly dispersed arrangement, disregarding the input data's clustering structure.

