# OpenReview forum: "Understanding Contrastive Learning through Variational Analysis and Neural Network Optimization Perspectives"
_ICLR.cc/2025/Conference — Submitted to ICLR 2025_

### Official Review · Reviewer_uzCb · 2024-10-25

**Soundness:** 3
**Presentation:** 3
**Contribution:** 2
**Rating:** 5
**Confidence:** 4

**Summary:**

This paper aims to understand the performance of contrastive learning (especially the property of preserving cluster structures). This paper first proposes that studying the global minimizer of the contrastive loss is insufficient since (within the invariance of the map T) there exist minimizers that is free of the data distribution. The authors further propose that the good properties of contrastive learning is related to the training dynamics and justify this via gradient flow in Theorem 4.2.

**Strengths:**

1. This paper is well written and organized, and offers very interesting insights into the understanding of contrastive learning.
2. The presented theoretical results are solid, which is also justified by experimental results.

**Weaknesses:**

1. While theoretical results in this paper are solid and insightful, my concern lies in the reasoning following Proposition 2.1. This result is based on the assumption on the existence of an invariant T, thus the minimizer to the optimization problem (on line 195) is implicitly depending on the distribution of data via the invariant T. In my understanding, this proposition is similar to a two-step analysis, i.e. fixing a map that captures the information of the underlying data and then by optimizing (line 195), the loss enforces some properties e.g. uniformity. Hence, I think Proposition 2.1 is insufficient to justify that the global solution of the contrastive loss cannot preserve the structures in data and the fine-grained analysis may be needed.
2. It would be more helpful if the theoretical results (e.g. Proposition 4.3) based on gradient flow can be elaborated with some concrete examples of \mu so that the gradients can be intuitively understood.

**Questions:**

1. [Notations] I assume equation (3) is referring to "equal almost surely".
2. [Prop 2.1] I was wondering even if the invariance of T is given, what should be the minimizer of the optimization problem (on line 195). Should the uniformity of the distribution be enforced by this loss? I think it could be relevant to the property of Gaussian potential.
3. [Choice of similarity measure] As the theory applies to a general family of the similarity measure \eta, I was wondering if the gradient flow would suggest the optimal choice of \eta? Or, why is the cross-entropy type of \eta better than the linear functions in practice.
4. [Explicit form of the gradient flow] Similar to what's mentioned in Weakness, It would be more helpful if the theoretical results (e.g. Proposition 4.3) based on gradient flow can be elaborated with some concrete examples of \mu so that the gradients can be intuitively understood. In addition, there are several theoretical analysis of contrastive loss and its global minimizers using concrete data distributions that can be relevant:
	- Ji, Wenlong, Zhun Deng, Ryumei Nakada, James Zou, and Linjun Zhang. "The power of contrast for feature learning: A theoretical analysis." Journal of Machine Learning Research 24, no. 330 (2023): 1-78.
	- Wang, Tongzhou, and Phillip Isola. "Understanding contrastive representation learning through alignment and uniformity on the hypersphere." In International conference on machine learning, pp. 9929-9939. PMLR, 2020.
	- Gui, Yu, Cong Ma, and Yiqiao Zhong. "Unraveling Projection Heads in Contrastive Learning: Insights from Expansion and Shrinkage." arXiv preprint arXiv:2306.03335 (2023).
	- Wen, Zixin, and Yuanzhi Li. "The mechanism of prediction head in non-contrastive self-supervised learning." Advances in Neural Information Processing Systems 35 (2022): 24794-24809.
5. [Figure 4] Are the arrays representing the direction of the gradients? In addition, on row 3, by saying vanilla gradient descent, what is the encoder for this setting?

I will be happy to raise my score if some fine-grained analysis of the gradient flow can be presented.

---

> ### Author Response · Authors · 2024-11-20
> **(Submission13238) Response to Reviewer uzCb**
>
> > C1: Regarding Prop 2.1
>
> We want to emphasize the constraint that T is invariant is in general completely independent of the data distribution \mu. This was not clear in our paper since the invariance condition in Eq.(3) involves \mu, but it can also be stated as x\in\R^D instead of x\sim\mu which we’ve revised the paper this way. For example, if the data is in \R^2 and the augmentations are to add noise in the y-coordinate of the data, then any map T(x,y) = S(x) that is independent of y is invariant, regardless of what the data distribution \mu is. The intention behind presenting Prop 2.1 is to highlight this kind of behavior. We’ve revised the paper to make this more clear.
>
> We would also like to clarify that Prop 2.1 is not the primary result of our paper; rather, it is a simple observation that can be easily derived from the contrastive learning loss function. One main goal of our paper is to show that the contrastive learning loss has infinitely many local minimizers, some of which may be independent of the data (as depicted in Thm 3.2). Another goal is to explain how NN optimization influences the training dynamics, guiding the optimization process toward a local minimizer that adheres to the data's geometric structure. This helps explain why contrastive learning algorithms yield such effective feature representations in practice.
>
> > C2: About concrete examples of \mu
>
> We agree that the material presented in Prop 4.3 and the subsequent theorem could be challenging to interpret. To address this, we have revised the paper by simplifying the NN formulation used earlier, resulting in clearer and more understandable results. Specifically, we replaced the weight matrix A in Eq(18) with a constant matrix. This change leads to simpler formulations in the subsequent theorems.
>
> To provide better insight into the kernel matrix, we introduced a new proposition following Prop 4.3 which examines the NN kernel matrix in the infinite-width limit, using a similar approach to the NTK framework.
>
> We also included a discussion of the NN gradient flow in the infinite-width limit. This not only simplifies the formulation but also enhances the interpretability of the NN's effects, making the overall arguments in the paper more accessible.
>
> > Q1: Notations
>
> We’ve revised.
>
> > Q2: Prop 2.1
>
> Given an invariant map T, there can be infinitely many local minimizers, each with the characteristics of both invariance and symmetry in the embedded distribution, as shown in Thm 3.2. Additionally, we indeed hypothesize that the minimizers and the corresponding loss values are dependent on the value of \tau in the Gaussian potential.
>
> To illustrate this, we examined the loss plot in Figure 3, where the loss is computed using the SimCLR function with an invariant feature map $T$ that maps to an evenly clustered distribution on the unit sphere. As shown, the loss plateaus after a certain point, indicating that beyond a specific number of clusters, the loss remains constant. Notably, the plateau point depends on $\tau$; as $\tau$ decreases, the threshold for the number of clusters at which the loss stops decreasing also decreases.
>
> > Q3: Choice of similarity measure
>
> The general form of the contrastive learning loss in Def 2.1 was presented to explore different choices for the similarity function $\eta$. While we focused on $\eta = e^{-|x-y|^2/\tau}$, we acknowledge that other choices for $\eta$ might lead to larger gradients, potentially accelerating optimization. This could also better explain the cross-entropy type of $\eta$, as the reviewer mentioned, but we did not explore this in depth.
>
> > Q4: Explicit form of the gradient flow
>
> In response, we have included all suggested references in the revised paper. We use a simple example where \mu represents a D-dimensional distribution, with data on the first d coordinates and noise along the remaining coordinates, ensuring orthogonality between the data and noise for easier analysis. Additionally, \mu is a clustered dataset with N clusters, each point being at a distance of \delta from the cluster center, simplifying the analysis.
>
> We agree that, despite the simplicity of this setup, the results—particularly the explicit form of the gradient flow—were challenging to interpret. We have replaced the NN with a simpler NN structure i.e. replacing $A$ within Eq(18) with a constant matrix, which we believe makes the results more accessible and easier to understand.
>
> > Q5: Figure 4
>
> You are correct: the arrows in the figure represent the negative of the gradients at each point. We have  included a description of this in the revised version for better clarity.
>
> The vanilla gradient descent does not require any encoder as the gradient descent step applies on the embedded points (Eq(14)). Specifically, the embedded points are updated based solely on the gradient of the contrastive learning loss, and this update does not involve the NN kernel, making it independent of the network structure.

---

### Official Review · Reviewer_ccng · 2024-11-03

**Soundness:** 4
**Presentation:** 3
**Contribution:** 2
**Rating:** 5
**Confidence:** 3

**Summary:**

This paper explores properties of self-supervised contrastive learning. In particular an optimality criterion for a generalized form of the popular NT-Xent loss function (used in the SimCLR framework) is derived and used to find a family of functions that satisfy this criterion.
Of particular interest is the fact that optimal solutions exist that are in some sense "independent" of the input data distributions, i.e. they do not recapitulate the clustering structure of the inputs.
The author's then utilize the NTK framework to analyze the inductive biases of training a neural network using said loss function.
This theoretical analysis of the learning dynamics and empirical results both shed light on how optimization of SSL loss functions with neural networks tend to find useful solutions/representations despite the fact that uninteresting optima exist.

**Strengths:**

- The core topic, why SSL leads to useful representations despite the fact that uninteresting minimizers exist, is of interest to the community, and utilizing NTK to examine the inductive biases of neural network training is an interesting approach.

- Including several different families of SSL (instance-contrastive/SimCLR, dimension-contrastive/VicReg, and non-contrastive/BYOL) in the analysis strengthens the generality of the analysis.

- The toy-setting and accompanying theoretical analyses are well described and useful for building intuition.

**Weaknesses:**

- The first main contribution, showing that SSL losses have minimizers that are in some sense independent of the input data distribution, is well known and there are multiple existing studies that propose theoretical reasons for why useful representations emerge from training (and I feel this works contribution and novelty could be strengthened by interfacing with them). Two examples of this type are:
  - "Understanding Contrastive Representation Learning through Alignment and Uniformity on the Hypersphere" [Wang and Isola, 2023]: shows that contrastive learning optimizes for embedding uniformity. Given this, and the fact that the loss is clearly invariant to the permutation of (invariant) embeddings it is obvious that minimizers from which there is no ability to separate ground truth classes.
  - "Provable Guarantees for Self-Supervised Deep Learning with Spectral Contrastive Loss" [HaoChen et al., 2022]: Defines a concept of the "augmentation graph," and provides bounds on downstream class separability given that class-membership clusters form connected sub-graphs. Similarly "Contrastive and Non-Contrastive Self-Supervised Learning Recover Global and Local Spectral Embedding Methods" [Balestriero et al., 2022] derive optimality conditions for various losses and give performance guarantees in the condition that augmentation similarity graphs overlap with class similarity graphs.

- It is not obvious to me what the implications of this theoretical work are for SSL practitioners. I believe the impact of the work would be strengthened substantially if the theory could be used to levy practical suggestions for improving the quality of learned representations (and strengthened even more if the author's could empirically demonstrate the efficacy of their suggestions!).

**Questions:**

- My understanding is that in the "Vanilla Gradient Descent" setting, the position of the embedding is directly optimized. My question is that, if the NT-Xent loss is used, how do you ensure that embeddings remain on the constraint surface (i.e. remain unit vectors)? Are embeddings projected to the unit sphere after each gradient step? Do the author's think there may be any implications for the results if some other way of remaining on the constraint surface (i.e. gradient steps that leave outputs on the unit sphere)?

- I wonder if the author's could comment on how their study (or its implications) are impacted by the near ubiquitous used of projector networks ("Guillotine Regularization: Why removing layers is needed to improve generalization in Self-Supervised Learning," [Bordes et al., 2022]) in SSL. Meaning most SSL methods actually discard the final layers of the network (where the loss is optimized) when transferring to downstream tasks. In light of this, might it be interesting to use a similar framework to analyze the dynamics of internal representations of deeper networks during SSL training (i.e. perhaps clustering structure is maintained longer or more pronounced in layers preceding loss evaluation)?

---

> ### Author Response · Authors · 2024-11-20
> **(Submission13238) Response to Reviewer ccng**
>
> > C1: Regarding invariance to the permutation
>
> We fully agree that it is rather obvious that there are minimizers of the loss that do not have the ability to separate ground truth data. This observation is mainly to motivate the main contribution of our work, which is to study the dynamics of training contrastive learning.
>
> Thank you for the reference; we were unaware of it but agree it’s highly relevant. In the revised paper, we've added a discussion on this and other related work. [Wang 2023] analyzes contrastive learning loss with invariant feature maps and the role of repulsive forces. They show that, asymptotically, the uniform distribution on the hypersphere is a local minimizer, leading to uniform embeddings.
>
> In our paper, however, we aim to demonstrate that there can be infinitely many local minimizers that satisfy invariance, while also adhering to a certain degree of symmetry in the latent distribution. This result does not imply that all minimizers are unrelated to the data distribution; rather, it suggests that many local minimizers can exist that are independent of the data. In practice, however, the computed solution from contrastive learning does indeed exhibit the geometric structure (e.g., clustering) of the data.
>
> The main question we address in our paper is: Why, among many local minimizers, does NN optimization consistently find a local minimizer that aligns with the data’s geometric structure? This specific question is not discussed in [Wang 2023].
>
> > C2: Regarding the graphs
>
> Thanks for pointing out these papers, which are indeed related to our work in some ways. The main assumption in both papers is that the augmentation graph is well-aligned with the class-membership graph; this assumption leads to guarantees for downstream tasks. This is very closely related to our notion of whether the encoder preserves any information about the source distribution; when it does this well, the graphs should be well-aligned and downstream tasks successful.
>
> We establish several results that are related to and interface with these works. First, we show that minimizing the contrastive loss alone does not guarantee that these graphs are well-aligned, since the encoder may destroy all the information about the source distribution. So in general there are no guarantees for downstream tasks. As the referee pointed out, some aspects of these results may be known.
>
> The main thrust of our paper is in studying the dynamics of NN optimization of the contrastive loss, which we show imparts some information from the source distribution into the encoded distribution. In some sense, this may also show that the class and augmentation graphs are well-aligned, leading to downstream guarantees, but this is beyond what we are currently able to show.
>
> > C3: Regarding the implication practical use
>
> This paper is primarily focused on theoretical analysis, aiming to understand why contrastive learning algorithms perform so well and demonstrating their state-of-the-art performance in the NN regime. As such our current results do not directly offer practical recommendations for improving existing state-of-the-art SSL models.
>
> > Q1: How are embeddings kept on the unit sphere
>
> For vanilla gradient descent, as you correctly pointed out, the optimization is performed using projected gradient descent. This is a reasonable choice in this context because the scaling of the norm of the embedded vectors (e.g., projection onto the unit sphere) does not affect the loss value, as the loss depends only on the angular distance. Furthermore, this is a straightforward approach in the unit sphere setting, as the process is nearly identical to Riemannian gradient descent, with the primary difference being the scale of the step size in the optimization.
>
> > Q2: Regarding the discarding of the last layer
>
> Indeed, in many contrastive learning methods, including SimCLR, the proposed architectures discard the final layers of the network when transferring to downstream tasks. Many studies suggest that discarding the last layer leads to better performance, and the representations obtained after discarding the final layer often show improved results.
>
> It certainly seems like this could be related to our work. If the encoder learned in practice shares some similarities with the “poor” minimizers we describe (that are independent of the source distribution), it may be that  this independence gradually strengthens with the depth of the network, and removing the final few layers could ameliorate this.
>
> This is just a hunch, but frankly, we do not have a clear theoretical insight into this phenomenon. In our analysis, we assumed a feature map parameterized by a NN without incorporating this step of discarding the final layer and we did not attempt to identify any theoretically grounded reasons for the effect of discarding the last layer. We agree this would be a very interesting direction for future work.

---

> > ### Comment · Reviewer_ccng · 2024-11-25
> >
> > Thank you to the author's for their response and clarifications. I agree with the author's that analyzing learning dynamics of contrastive losses is an interesting direction which has the potential to deliver complimentary insights to existing theoretical explorations of SSL representation learning. However I think the eventual impact of this paper will hinge on it's ability to apply/adapt the methods developed here to provide insights in a setting that is closer to those used in practice (in terms of network architecture and dataset complexity).
> >
> > I opt to keep my score, and encourage the author's to build on their observations in at least one of the interesting directions mentioned above for the resubmission.

---

> > > ### Author Response · Authors · 2024-12-02
> > >
> > > Thank you so much for your insightful comments on the practical implications of our work. We greatly appreciate your thoughtful feedback.
> > >
> > > To clarify, one of the main practical implications of the paper is that the paper provides better understanding of how neural network optimization for contrastive learning effectively guides the model towards a desirable local minimizer. This minimizer is aligned with the data's geometric structure, among the many local minimizers that could be independent of the data's structure. Furthermore, as indicated in Theorem 4.2, the paper suggests that neural network optimization on the contrastive learning loss deviates from perfect invariance. In practice, this means that the optimization process may guide the model to stationary points that do not fully satisfy the invariance defined in Equation (3).
> > >
> > > Thus, the practical contribution of this paper is to provide a deeper understanding of what state-of-the-art contrastive learning algorithms are actually doing, and the kinds of solutions they are likely to find. Again, thank you for your valuable input, which has helped us clarify these aspects.

---

### Official Review · Reviewer_Y5iN · 2024-11-03

**Soundness:** 1
**Presentation:** 1
**Contribution:** 1
**Rating:** 3
**Confidence:** 5

**Summary:**

This study presents preliminary results investigating the SimCLR contrastive learning method to better understand its effectiveness. The findings suggest that the SimCLR loss function alone is insufficient for selecting an optimal minimizer, and a thorough analysis of the neural network training dynamics is essential.

**Strengths:**

This study examines the learning dynamics of the SimCLR loss function.

**Weaknesses:**

The motivation for the study in the introduction is unclear. There is a lack of a comprehensive overview of the numerous analytical studies aimed at understanding and improving the success of contrastive learning. It is essential to explicitly outline how this research differs from existing studies and what new findings it presents. A detailed section on related works should be included to address these points. Additionally, it should be clarified how this research serves as a complement to existing studies on dimensional collapse.

The two contributions claimed in this research lack sufficient evidence to support their relevance to the success of SimCLR. In fact, the results appear to highlight the shortcomings of contrastive learning rather than reinforcing its effectiveness. Clear justification is needed to demonstrate how these contributions substantiate the efficacy of contrastive learning.

Furthermore, the mathematical notation used in this research is not conventional, which diminishes readability. For instance, many subsequent studies, including those on SimCLR, commonly use T for augmentation and f for the encoder. In contrast, this study employs notations that are inverse to this convention.

Assumptions necessary for the validity of Equation 3 have been omitted. The absence of these assumptions may lead to an overgeneralization of the theoretical results derived from them. For example, if the augmentation is exceedingly strong—such as an augmentation that reduces all values to zero—Equation 3 would not hold true. Moreover, during the actual training of contrastive learning, the learning occurs in batches, further complicating the validity of Equation 3.

**Questions:**

The primary objective of contrastive learning is to encode various features present in the data into a representation. While clustering-related features are indeed included, there are numerous other features that also contribute to this encoding process. Given this consideration, the assumption in this study that clean data is highly structured or clustered may be viewed as a narrow and potentially unrealistic constraint.

---

> ### Author Response · Authors · 2024-11-20
> **(Submission13238) Response to Reviewer Y5iN**
>
> ## Response to the comments about weaknesses
>
> > **C1: The motivation for the study in the introduction is unclear.**
>
> Several other referees have pointed out related papers that we were not aware of. In the revision, we have added a significant amount of additional discussion citing and connecting our results to previous work. In the original manuscript we cited several works on dimensional collapse, but these are very much orthogonal to our work (the issues we raise are independent of collapse).
>
> > **C2: The two contributions claimed in this research lack sufficient evidence.**
>
> We would like to clarify that our paper is not intended to highlight shortcomings of contrastive learning. The main motivation of our study is exactly to understand the effectiveness of contrastive learning. We have used it in downstream tasks in some of our other work and we agree it is extremely effective; we wanted to understand why.
>
> We describe our main motivation in the introduction, and illustrate it in Figure 1, which shows how contrastive learning can successfully uncover rich and interesting cluster structure in Cifar-10 that is not present in the pixel space directly.
>
> Our initial ideas in this work were to study the loss function for contrastive learning, but we quickly realized that the loss function alone does not explain this effectiveness (e.g., it cannot explain Figure 1) since minimizers of the loss need not be related to the distribution of the data in any way. Another referee agreed with this assessment, and pointed out that it has appeared before in previous work that we were not aware of (though it is not the main point in our paper).
>
> The main novel contribution of our paper is a set of initial results that aim to understand the effectiveness of contrastive learning through the dynamics of optimization/training when the embedding is parameterized by a neural network. Essentially the punch line is that the training dynamics bias contrastive learning towards “selecting” good approximate minimizers of the loss function, instead of poor ones that are distributionally independent of source data. As far as we are aware, this is a new point of view that has not been observed in the literature.
>
> > **C3: Mathematical notation used in this research is not conventional.**
>
> We apologize for using different notation than is standard. In the revision we have switched the notation so that T is the encoder and f is the augmentation.
>
> > **C4: Assumptions necessary for the validity of Equation 3 have been omitted**
>
> We believe this is just a misunderstanding due to our poor choice of notation (swapping T and f). Equation 3 is simply the invariance condition for the encoder, which is also called “alignment” in some previous works. Achieving Equation 3 is one of the main goals of contrastive learning and has been studied/assumed by many previous works. In the example the reviewer gave, if the augmentation is indeed exceedingly strong and reduces all values to zero (dimensional collapse), then Equation 3 trivially holds (i.e., dimensional collapse helps with invariance).
>
> ## Response to questions
>
> > **Q1: The primary objective of contrastive learning is to encode various features present in the data into a representation. While clustering-related features are indeed included, there are numerous other features that also contribute to this encoding process. Given this consideration, the assumption in this study that clean data is highly structured or clustered may be viewed as a narrow and potentially unrealistic constraint.**
>
> We disagree with this point. It is not just clustering features we are studying, it is any features that are related to the source distribution of the data. If the encoded distribution is independent of the source distribution, then the encoding can in general be useless for all downstream tasks. As another referee pointed out, the loss allows for arbitrary permutation of the data points in the encoder, which is akin to randomly corrupting all the labels in a downstream classification task.
>
> We are not claiming that contrastive learning produces pathological solutions like this in practice; the point is that the loss function allows for it and we would like to understand why these pathological minimizers do not arise in practice.

---

> ### Comment · Reviewer_Y5iN · 2024-11-26
>
> **C1:** The revised manuscript still lacks an overview of the numerous analytical studies that focus on understanding and enhancing the success of contrastive learning. It is strongly recommended that the authors include a dedicated "Related Work" section. This would allow for a clearer comparison of how the current research differs from existing studies and highlight the novel contributions of this work.
> Additionally, if this study is independent of research on dimensional collapse, the rationale for dedicating a significant portion of the manuscript to collapse-related studies—those that do not seem directly relevant to the current research—requires further justification.
>
> **C2:** The main contribution outlined in the authors' rebuttal is not clearly articulated in the manuscript itself. It is essential to clearly define the contribution and ensure that it is well-supported by the content of the paper.
>
> **C3:** Thank you for the updated version; it has enhanced the readability of the manuscript.
>
> **C4:** To be more rigorous, the statement "Achieving Equation 3 is one of the main goals of contrastive learning and has been studied/assumed by many previous works." is inaccurate. The primary objective of contrastive learning is to maximize the "cosine similarity" in Equation 2 between positive pairs. Moreover, in practice, the cosine similarity between positive pairs does not reach 1, even when using weak augmentation functions. Therefore, the equation 3 should be revised rigorously to include appropriate assumptions.
>
> **Q1:** Contrary to the authors' claim, the study does not provide comprehensive results beyond clustering. It is strongly recommended that the authors incorporate additional empirical examples that extend beyond clustering-related features to strengthen the validity of the study.
>
> The authors' rebuttal does not adequately address many of the concerns raised. Significant revisions are still required to improve the manuscript. As a result, I am maintaining my original score.

---

> > ### Author Response · Authors · 2024-12-02
> >
> > Thanks for your additional comments. We must say some of them are confusing to us, and do not match the actual content in our paper.
> >
> > **C1**: It is true we do not have a section explicitly entitled "related works". We thought it worked better to cite and describe the works within the body of the text when they were relevant. In the future we will add a section with this explicit title. It would have been much more helpful if the referee could suggest even 1 (or 2) papers, instead of just insisting the related works section is missing. The other referees were all extremely helpful in this regard, and we have cited and discussed all the papers they suggested in our revision.
> >
> > The reviewer also says that we spend a "significant portion" of the manuscript on dimension collapse. This is simply not true. There are a handful of sentences in the introduction mentioning it and saying it is orthogonal to our work.
> >
> > **C2**: We respectfully point out that there are two bulleted points in the introduction (pages 2/3) that clearly outline the contributions, as described in our response.
> >
> > **C3**: We were more than happy to do this.
> >
> > **C4**: This also seems to be a strange comment; the referee is simply explicitly stating the loss function in contrastive learning. It is like saying that the goal of classification is to minimize the cross entropy between the soft-max output of the network and the target labels, when the ultimate goal is to correctly classify images. The contrastive loss is clearly encouraging the map to be invariant, and other works have taken this point of view before (e.g., the citations provided by the other referees). We agree that the minimizers in practice may not become exactly invariant, but this is just a simplifying assumption to make the analysis tractable.
> >
> > **Q1**: This is also not true. It is only the last section 4.2 that studies a clustered data set. The rest of the results do not assume this setting. But we must say that, even if the whole paper were devoted to the clustering ability of contrastive learning, this would seem like a completely worthy objective for an ICML conference paper.

---

### Official Review · Reviewer_LPiL · 2024-11-04

**Soundness:** 3
**Presentation:** 3
**Contribution:** 3
**Rating:** 6
**Confidence:** 3

**Summary:**

The paper proposes to study trivial minimizers of InfoNCE, their link to the data distribution as well as their appearance when using neural networks. Indeed, the InfoNCE criterion has trivial minimizers (collapsing to a single point, or a discrete measure) that can be learned by gradient descent. Studying neural network dynamics, the authors show that the clustering structure of the data can remain present during training, and may help explain the success of contrastive self-supervised learning.

**Strengths:**

- The focus on the training dynamics with a neural network leads to an analysis that is significantly closer to practice than what an analysis of the loss function alone would provide. While it is common knowledge that trivial solutions exist, it is important to understand how they emerge or not
- The study of the impact of the data distribution, while preliminary, sheds light into the success of SSL through the preservation of properties of the data distribution (e.g. clustering here)
- The tools used in this work could be repurposed to study in more depth the influence of the data distribution on the representations learned via SSL. Notably, the brittleness of methods to non-uniform data distributions that is observed empirically may be theoretically explainable.

**Weaknesses:**

- The caption of Figure 2 should be self-sufficient, describing every part of the figure. Notably, describing that $T$ is the embedding function, $f$ corresponds to the augmentations etc would make the plot more self-contained and help provide a more graphical description of the setup.
- Most of the analysis focuses on the setting where $T$ is already invariant to $\nu$. This means that the studied criterion is reduced to the
repulsive force rather than the complete InfoNCE. A reference to [1] is also missing, which studied minimizers of InfoNCE, and notably the link between minimizing the repulsive force and the uniform distribution on the hypersphere.

[1] Wang, T., & Isola, P.. Understanding contrastive representation learning through alignment and uniformity on the hypersphere.ICML, 2020.

**Questions:**

- In figure 4, why is the choice of number of iterations for the two rows different ? Is it simply for visualization (to get similar “convergence” states), or does the behaviour change if we continue the first row to 1000 iterations ? I would suggest explaining the rationale behind the choices or show plots at convergence as well.
- More of an open ended discussion, but how do you think that the choice of architecture for the encoder influences how the underlying structure of the data is present in the embedding ? We see in figure 4/5 that with a 1/4 layer neural network, we roughly keep the clustered structure of the data (except the blue cluster which is split in two at 200 iterations). Perhaps more powerful architectures (ResNet,ViT) may have behaviours akin to the “vanilla gradient descent” setting, as asymptotically, a network with infinite capacity would be closer to this setting. Experiments are of course not expected here.
- Previous work [2] studied the importance of uniform clusters of features in self-supervised learning, mainly how the learned representation better capture  features present uniformly rather than following a power law. Perhaps some of the analysis presented here could be extended to a setting with non uniform clusters to help explain why contrastive learning tends to fail on non-uniformly distributed data. Again, this is mainly a discussion point.

[2] Assran, M., Balestriero, R., Duval, Q., Bordes, F., Misra, I., Bojanowski, P., ... & Ballas, N. (2022). The hidden uniform cluster prior in self-supervised learning. arXiv preprint arXiv:2210.07277.

---

> ### Author Response · Authors · 2024-11-20
> **(Submission13238) Response to Reviewer LPiL**
>
> ## Response to the comments about weaknesses
>  > **Comment about Figure 2**
>
> We’ve improved the caption of Figure 2 based on your comment.
>
> > **Comment about invariance**
>
> Thank you for your comment on the analysis and the recommendation of the reference. The reviewer’s observation that the analysis focuses on the invariant setting is correct. This is intended to describe the local minimizers of the InfoNCE loss, and the paper shows local minimizers that satisfy the characteristic of being invariant to $\nu$. We have included the suggested reference and added more details about the reference, as it also discusses how the repulsion force leads to minimizers that are uniformly distributed on a sphere. We will clarify the differences between our work and the suggested reference.
>
> To highlight the main difference, our paper examines the Euler-Lagrange equation of the variational problem and demonstrates the general form of the local minimizers of the InfoNCE loss, where the uniform distribution is one aspect. It provides a characterization of these local minimizers in a more general sense, thus improving upon the results presented in the suggested reference. Thank you for your thoughtful comment. While the paper [1] demonstrates that the uniform distribution satisfies the local minimizer asymptotically, our result is more general. Specifically, Theorem 3.2 is formulated as an expectation over the density, where the density can be general (e.g. either discrete or continuous), thereby encompassing the asymptotic result naturally. Furthermore, our theorem provides a broader characterization of local minimizers, which includes the uniform distribution as a special case.
>
> As illustrated in the loss plot in Figure 3, which shows the contrastive learning loss for an invariant feature map that maps to a latent distribution with an evenly distributed discrete measure and a specified number of clusters, we observe that the contrastive learning loss stops changing after a certain number of clusters. This suggests that all these clusters share the same loss for a given value of $\tau$, indicating that they are also valid local minimizers, in addition to the uniform distribution.
>
> We hope this clarifies the distinction and provides further insight into the generality of our results. Thank you again for your valuable feedback.
>
> ## Response to the comments about questions
>
> > **Q1: In figure 4, why is the choice of number of iterations for the two rows different?**
>
> Thanks for the question. The iterations were chosen to best illustrate the training dynamics for both cases: with and without neural network optimization. The inclusion of the neural network kernel matrix in the gradient descent process significantly alters the training dynamics, making them much different from the case without neural networks. This is why the iterations were selected differently for each case. We stopped at 200 iterations because the dynamics do not change significantly beyond that point. We will explain this in more detail in the paper, and the added information can be found in the revised version.
>
> > **Q2: Regarding the choice of architecture for the encoder**
>
> This is an excellent question. So far, we have tested relatively simple neural networks, including fully connected and convolutional networks with multiple layers ranging from 2 to 10, and we observed the same behavior. The outcome of considering more complex neural network structures, or the case of infinite width or depth of neural networks, is not yet known. This is a very interesting research direction, and we plan to study it in the future.
>
> We expect that our observations are in some way related to the frequency bias of training dynamics to low frequencies, which has been established even for large scale deeper networks, so we tentatively expect similar behavior for other architectures.
>
> > **Q3: Regarding the non uniform clusters.**
>
> Thank you for the insightful feedback on the paper and the very useful reference. This is indeed a very interesting direction, and we’ve updated the last section of the paper to consider a more general clustering structure, rather than the uniform clusters originally discussed. By doing so, we have revised Theorem 4.4 to better illustrate the effect of cluster imbalance on the gradient flow. As shown in the revised version of the paper, the gradient is scaled according to the ratio of the sizes of the clusters within the dataset. If one particular cluster has a smaller number of points and another has a larger number, it is possible that the gradient from the smaller cluster could be negligible compared to the larger cluster. This is a preliminary result and does not fully explain the phenomenon discussed in [2], but we aim to address it further in future work.

---

> > ### Comment · Reviewer_LPiL · 2024-11-25
> >
> > Thank you for the detailed response and the clarifications added to the paper. However, I want to remind the authors that the revision must adhere to the 10 page limit (same as for the original submission), so I'd recommend shortening the paper to fit in this limit while it's still possible.
> >
> > Overall, while I believe that the paper makes worthwhile contributions to the understanding of self-supervised learning, the considered setup where the invariant map is taken for granted limits the impact of the work. I will thus keep my original score for now.

---

> > > ### Author Response · Authors · 2024-12-02
> > > **Response to Reviewer LPiL**
> > >
> > > Thank you very much for your insightful comments. Your feedback has been incredibly helpful and has enabled us to make meaningful improvements to the paper. As per your suggestion, we have uploaded the revised version of the paper, now within the 10-page limit.

---

### Meta-Review · Area_Chair_4Lfw · 2024-12-19

**Metareview:**

This paper conducts a mathematical analysis of the SimCLR method, which aims to better understanding the geometric properties of the learned latent distribution. After the response, it receives mixed ratings, including three rejects, and one accept. The advantages, including the training dynamics analysis and good presentation, are recognized by the reviewers. However, they are also concerned about the assumption, undesirable overview, lack of concrete examples, etc. I also think the current manuscript does not meet the requirements of this top conference. I suggest the authors carefully revise the paper and submit it to another relevant venue.

**Additional Comments On Reviewer Discussion:**

After the discussion, most of the concerns still exist. All reviewers keep their original ratings unchanged.

---

### Decision · Program_Chairs · 2025-01-22

Reject